# Structures of Atm1 provide insight into [2Fe-2S] cluster export from mitochondria

Ping Li[1], Amber L. Hendricks[2], Yong Wang[3,4], Rhiza Lyne E. Villones[5], Karin Lindkvist-Petersson [1], Gabriele Meloni [5], J. A. Cowan[2], Kaituo Wang [6] ✉ & Pontus Gourdon [1,6] ✉

In eukaryotes, iron-sulfur clusters are essential cofactors for numerous physiological processes, but these clusters are primarily biosynthesized in mitochondria. Previous studies suggest mitochondrial ABCB7-type exporters are involved in maturation of cytosolic iron-sulfur proteins. However, the molecular mechanism for how the ABCB7-type exporters participate in this process remains elusive. Here, we report a series of cryo-electron microscopy structures of a eukaryotic homolog of human ABCB7, CtAtm1, determined at average resolutions ranging from 2.8 to 3.2 Å, complemented by functional characterization and molecular docking in silico. We propose that CtAtm1 accepts delivery from glutathione-complexed iron-sulfur clusters. A partially occluded state links cargo-binding to residues at the mitochondrial matrix interface that line a positively charged cavity, while the binding region becomes internalized and is partially divided in an early occluded state. Collectively, our findings substantially increase the understanding of the transport mechanism of eukaryotic ABCB7-type proteins.

The ABCB7/HMT1/ABCB6 family of adenosine triphosphate (ATP)-binding cassette (ABC) proteins also known as ABC transporters of mitochondria 1 (Atm1)[1,2] is present in essentially all eukaryotes, spanning the membranes surrounding the cells and organelles such as mitochondria, Golgi and lysosomes, as well as in the envelopes of certain prokaryotes. Physiologically, in prokaryotes, they are associated with heavy metal homeostasis, as shown for the homolog NaAtm1 from the alphaproteobacterium *Novosphingobium aromaticivorans*[3]. In contrast, in eukaryotes, the well-studied ScAtm1 from the yeast *Saccharomyces cerevisiae* is required for the occurrence of iron–sulfur (Fe–S) clusters in cytosolic proteins[4,5] and tRNA thiolation[6]. Disruption of Atm1 causes iron accumulation in mitochondria[7,8]. In humans, mutations of ABCB7, for example, result in the accumulation of granular iron in mitochondria surrounding the

nucleus (sideroblast) and the severe disease X-linked sideroblastic anemia, which prevents red blood cells to mature from the erythroblast state[9]. In these cases, there is a deficiency of cytosolic Fe–S clusters that impair protoporphyrin biosynthesis and the subsequent formation of heme necessary for the formation of normal erythrocytes[10].

At the molecular level, Atm1 proteins have been implicated in the transport of a wide variety of molecular cargo across membranes at the expense of ATP, through nucleotide hydrolysis facilitated by a conserved catalytic glutamate residue[2,11,12]. In prokaryotes, Atm1 has been implicated in the export of heavy metals[3]. In contrast, in eukaryotes, the function appears related to iron homeostasis, to the reconstitution of cytosolic Fe–S cluster proteins, and in particular to the upstream transfer of Fe–S clusters[4,7], which are generated in mitochondria but

[1]Department of Experimental Medical Science, Lund University, Sölvegatan 19, SE-221 84 Lund, Sweden. [2]Department of Chemistry and Biochemistry, The Ohio State University, 100 West 18th Avenue, Columbus, OH 43210, USA. [3]Institute of Quantitative Biology, College of Life Sciences, Zhejiang University, Hangzhou 310027, China. [4]The Provincial International Science and Technology Cooperation Base on Engineering Biology, International Campus of Zhejiang University, Haining 314400, China. [5]Department of Chemistry and Biochemistry, The University of Texas at Dallas, 800 W Campbell Rd., Richardson, TX 75080, USA. [6]Department of Biomedical Sciences, Copenhagen University, Maersk Tower 7-9, Nørre Allé 14, DK-2200 Copenhagen N, Denmark. ✉e-mail: kaituo@sund.ku.dk; pontus.gourdon@med.lu.se

are incorporated into proteins in the cytoplasm via the cytosolic iron–sulfur protein assembly machinery (CIA)[13,14]. Importantly, glutathione (GSH), a thiol-containing Glu-Cys-Gly tripeptide available in cells at concentrations up to 10 mM, represents a key component of the molecular species that is a cargo molecule for Atm1 proteins[15]. Specifically, the transport activity of both prokaryotic and eukaryotic members is stimulated by the presence of GSH or its oxidized version (GSSG)[3,16]. Furthermore, both yeast and human Atm1 have been shown to permit passage across cellular membranes of tetrameric "macrocyclic" glutathione-coordinated Fe–S clusters, abbreviated [2Fe-2S](GS)4[17–19] (Supplementary Fig. 1). The latter is of interest because findings have indicated such complexes are stable in aqueous media thereby permitting [2Fe-2S] clusters to be mobilized in the cell without a protein scaffold[18–21]. In addition, [2Fe-2S](GS)$_4$ cofactors are biosynthetically accessible via glutathione extraction from the ISU scaffold protein[18], and can reconstitute certain apo iron–sulfur cluster proteins, and represent a plausible component of the cellular labile iron pool[17,20,22,23].

While an alternative cargo of a polysulfide adduct of oxidized glutathione has been suggested from studies of the Atm3 from the plant *Arabidopsis thaliana* and Atm1 from *S. cerevisiae*[24,25], the physiological relevance of this species and how it relates to Fe–S cluster assembly has not been elucidated[24,25]. Such glutathione-based cargo may have relevance in the context of tRNA thiolation, but additional work on the molecular details is required. With regard to Fe–S cluster trafficking, an iron-containing cargo appears most likely. Indeed, biochemical characterization of the yeast transporter supports a Fe–S type cargo molecule, and a mitochondrial origin for cytosolic [2Fe-2S] centers, with potential sulfur-based intermediates directed to cytosolic tRNA thiolation[24,25].

Structural information on Atm1 proteins is limited to a single state of human ABCB7, hABCB7[26], as well as conformations of orthologs from the yeast *S. cerevisiae*, ScAtm1[27,28], from *A. thaliana*, AtAtm3 (another of the three Atm orthologues in *Arabidopsis*)[29], and from a member from the *N. aromaticivorans*, NaAtm1[3,30]. The latter species is of interest as the phylum underpins the endosymbiotic model of how mitochondria and chloroplasts were engulfed by early prokaryotic cells to generate eukaryotes. The studies have revealed an overall homodimer architecture similar to other ABC transporters, with four main parts; two transmembrane domains (TM-domain) consisting of alpha helices that come into contact with the cargo, and two nucleotide-binding domains (NBDs) where ATP binds and is hydrolyzed to ADP[12] (Supplementary Fig. 2a). Each monomer harbors six transmembrane helices (TM1–TM6), with TM4 and TM5 wrapping around the other monomer, thereby asserting the dimer assembly. While the NBDs are C-terminal to the TM-domain, connections between the domains are also established via the TM2/TM3 and, in particular, the TM4/TM5 hairpins. Collectively, the available structural and biochemical data point toward Atm1 proteins exploiting the classical switch mechanism for ABC transporters, with coupled movements of the NBDs and the TM-domains, thereby allowing cargo to cross the membrane[12]. According to this model, the protein is initially in an open inward-facing state where no nucleotide or cargo is bound with well-separated NBDs. The transported molecule and ATP then associate with the TM-domain and the NBD, respectively, triggering conformational changes of the NBDs and the TM-domains, resulting in an occluded structure with enclosed cargo in the TM-domain and with the NBDs in close proximity to each other. Then, in the outward-facing open state, the cargo is released. Finally, ATP hydrolysis takes place and upon release of inorganic phosphate and ADP, the protein returns to its original open, inward-facing orientation via a closed configuration (Fig. 1a)[12,31].

Despite the insight regarding the overall transport cycle, the molecular determinants that govern transport and selectivity in Atm1 proteins remain enigmatic. In prokaryotes, structural and functional

studies of NaAtm1 have implicated a direct role in heavy metal detoxification[3]. In eukaryotes, previous studies have suggested that Atm1 transports sulfur intermediates required for cytoplasmic tRNA thiolation[24,25]. For Fe–S cluster containing proteins, glutathione-bound iron–sulfur clusters, [2Fe-2S](GS$_4$), represent the most likely cargo to provide clusters for cytosolic proteins. This is supported by the observations that GSH depletion affects the activity of cytosolic iron–sulfur proteins[24], and that Fe–S cargo facilitates CIA-type cluster assembly[25]. Because these clusters have been shown to cause a significant increase in the ATPase activity of Atm1[18], are actively transported in proteoliposome Atm1 models[19], and their relative solubility and link to [2Fe-2S] protein biogenesis and iron homeostasis as outlined above, they are deemed likely precursors for cellular protein–cluster reconstitution[17–20,22,23].

The poorly understood mechanism of mitochondrial export relates to the absence of structural information, in particular cargo-bound forms of eukaryotic members that are associated with the Fe–S clusters. Structures of eukaryotic Atm1 proteins have primarily been stabilized in two inward-facing configurations; namely, an inward-open and an occluded state, the former with and without bound GSH or GSSG[26–29]. These revealed a highly electropositive uptake region formed by TM3, TM5, and TM6[27], without elucidating further details regarding cargo transport, as it has only been observed in the inward-open configuration. In contrast, in the case of the bacterial members, the cargo-uptake cavity in NaAtm1 is considerably more electroneutral, but instead, heavy metal glutathione adducts have been revealed[3]. Inward-facing occluded structures have demonstrated two adjacent GSSG (or derivatives, including a Hg-bound form) pockets 5 Å apart close to the membrane interface, on separate sides of the GSH identified in the eukaryotic structure[3,27]. However, other recovered states of NaAtm1 have not demonstrated the presence of any transported cargo, but pockets compatible with GSH-facilitated transport have been identified, occasionally with weak indications of additional density[30]. Consequently, the molecular bases underlying heavy metal and Fe–S cluster transporting bacterial and eukaryotic members, respectively, are debated, and it is unknown whether similar mechanistic principles apply between these subclasses of Atm1 transporters.

By use of an ortholog from thermophilic filamentous fungus *Chaetomium thermophilum*, herein we report an investigation of glutathione-complexed [2Fe-2S] cluster cargo transport mediated by eukaryotic Atm1 proteins, combining structural studies with functional characterization efforts. We report multiple intermediates of the transport process and define structural models for binding of [2Fe-2S](GS)$_4$ cluster complexes to an Atm1 transporter, pinpointing molecular determinants that separate the prokaryotic and eukaryotic subclasses. Aggregated, our findings provide a framework for increasing understanding of the Atm1 transport mechanism, for analysis of missense mutations in human ABCB7, and for downstream rational drug-design efforts.

## Results and discussion
### Overall structure
To shed further light on the transport mechanism of eukaryotic Atm1 proteins we selected a member from *C. thermophilum*, CtAtm1, an organism that has shown great promise as a model organism for structural biology due to the stability of its proteins at high temperatures and its eukaryotic status[32,33]. CtAtm1 was isolated from *Escherichia coli*, purified in the detergent n-dodecyl-β-D-maltopyranoside (DDM), and successfully reconstituted into lipid nanodiscs (Fig. 1b). The recovered protein sample exhibited clear and comparably robust ATP hydrolysis activity in detergent solution and in the disc-configuration without the addition of exogenous cargo (Supplementary Fig. 2c). In contrast, the E603Q dead-mutation, targeting the invariant catalytic glutamate of the NBD, rendered the protein inactive

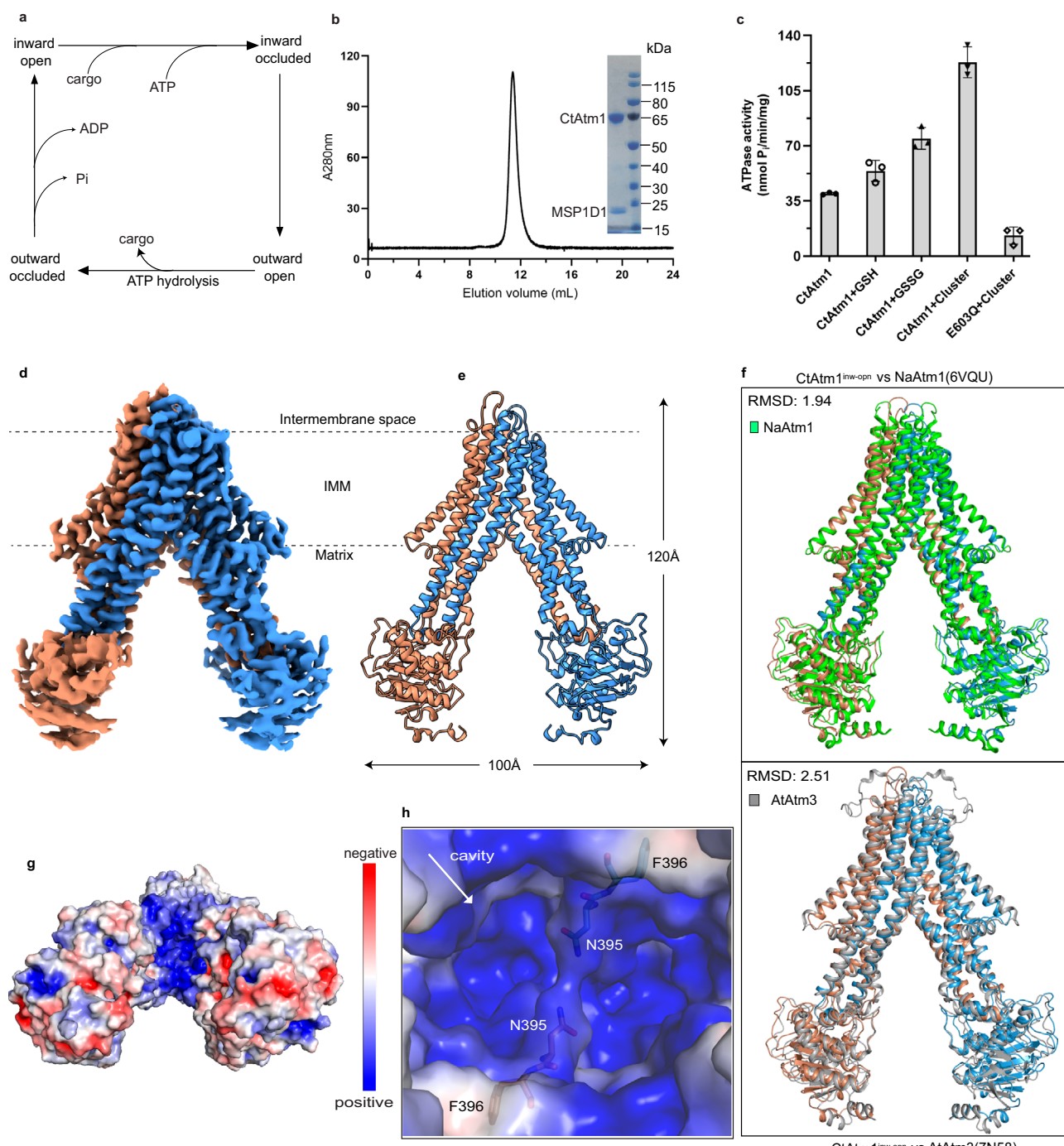

**Fig. 1 | Transport cycle, function, and inward-open structure. a** Atm1-type ABC proteins presumably exploit a transport mechanism that includes inward-facing, occluded, outward-facing, and closed conformations, and requires ATP and cargo to turnover. **b** CtAtm1-discs size-exclusion chromatography profile and associated Coomassie-stained SDS-PAGE analysis (nanodisc reconstitutions were performed more than three independent times). **c** ATPase activity of CtAtm1 in detergent solution and 20 μM GSH, GSSG, or cluster. Data points represent the means of three independent measurements and error bars indicate standard deviation. **d** Cryo-EM map of CtAtm1 in the inward-facing open conformation, CtAtm1[inw-opn], with one monomer shown in blue and another in brown. **e** The structure of CtAtm1[inw-opn] with 12 transmembrane helices and two NBDs. **f** Structural comparison of CtAtm1[inw-opn] (blue and brown) with the equivalent conformation of NaAtm1 colored green (PDB-ID 6VQU, green) and AtAtm3 colored gray (PDB-ID 7N58, gray). **g** Surface charge of CtAtm1[inw-opn] as observed from the mitochondrial matrix. **h** Close-view of the positively charged cavity with residue pairs Asn395 and Phe396 (same orientation as in panel **g**).

(Supplementary Figs. 2c & 3). As previously shown for protein ScAtm1[31] and hABCB7[34], the ATPase activity of wild-type CtAtm1 was stimulated by the presence of either GSH, GSSG or [2Fe-2S](GS)$_4$ clusters, the latter synthesized as previously reported[17–19]. Hence, these three compounds can all trigger function. However, the turnover was higher in the presence of the cluster, relative to GSH or GSSG (Fig. 1c). Next,

we determined a single particle cryo-EM structure at an average resolution of 3.0 Å, CtAtm1[inw-opn], exploiting wild-type protein without supplementation of cargo but in the presence of ATP (Fig. 1d, e, Supplementary Fig. 4, and Methods). The computed maps display well-resolved TM-domains, while the resolution of the NBDs is somewhat lower, and hence we are unable to confirm the absence of nucleotide

binding (Fig. 1d, Supplementary Fig. 5a, and Supplementary Videos 1–3). Nevertheless, de novo building of almost the entire structure was possible (580 modeled amino acids), with the exception of the first 94 residues in the N-terminus that includes the signal sequence for mitochondrial localization, and the last 26 residues of the C-terminus, likely related to flexibility of both termini (Fig. 1d). As anticipated, the architecture of CtAtm1 harbors the classical ABC-fold, packing into a homodimer with six transmembrane segments (TM1–TM6) in each monomer (Fig. 1e and Supplementary Fig. 2b).

The isolated structure adopts an inward-facing open conformation without obvious connection between the NBDs, homologous to a previously determined state of NaAtm1 (PDB-ID 6VQU[30], RMSD 1.9 Å) and AtAtm3 (PDB-ID 7N58[29], RMSD 2.5 Å), which we interpret as a pre-translocation arrangement (Fig. 1f). Thus, the configuration is distinct from the available structure of eukaryotic ScAtm1 and hABCB7 (PDB-ID 4MYC[27], 7PSL[28] and 7VGF[26], RMSD 4.2, 3.8, and 3.2 Å), in which the NBDs interact via C-terminal extended helices in the *S. cerevisiae* structures (Supplementary Fig. 6). As such, our structure is more open, exposing a large cavity (>7000 Å³ as calculated using the software CastP3.0 with a 2.5 Å radius probe[35]) that spans more than half of the membrane into the TM-domain (Fig. 1g, h). The cleft is decorated throughout with electropositive amino acids and is partially blocked at several levels: (1) at the matrix-membrane interface through the Asn395 pair of TM6 (one per monomer, conserved among eukaryotic Atm1, Supplementary Fig. 3), establishing an "inner" gate (the first from the mitochondrial matrix), and additional residue pairs closer toward the mitochondrial intermembrane space that firmly links the TM6 pairs (Fig. 1h and Supplementary Fig. 7). Notably, these amino acids also divide the cavity from Asn395 into two separate, symmetrically related, parts. In contrast to the overall arrangement, these properties are similar to the equivalent cavity of ScAtm1, AtAtm3, and hABCB7, while the NaAtm1 pocket is less electropositive, features that likely are key to understanding how different cargo is transported by eukaryotic and prokaryotic Atm1 proteins, respectively (Supplementary Fig. 8a, b).

### Glutathione-complexed [2Fe-2S] uptake

To understand how glutathione-complexed [2Fe-2S] clusters are recruited to Atm1 proteins, we subsequently sought to isolate a structure in the presence of cargo. To capture an early protein–cluster complex intermediate, we first supplemented wild-type protein with the cluster and the non-hydrolysable nucleotide analog adenosine 5′-β,γ-imidoadenosine triphosphate (AMP-PNP), but only inward-facing structures were recovered (Supplementary Fig. 9). Next, we exploited the E603Q form also for the structural studies, as this substitution in NaAtm1 has been shown to stabilize an occluded conformation[30]. However, the addition of ATP and [2Fe-2S](GS)₄ precipitated the protein. Subsequently, the E603Q sample was instead supplemented with AMP-PNP and the cluster (Methods). Based on this setup, three separate cryo-EM maps were calculated to an overall resolution of 3.2, 2.9, and 3.0 Å, respectively (Supplementary Fig. 10). The latter two yielded highly homologous structures to our determined CtAtm1$^{inw-opn}$ structure, including poorly resolved NBDs (Supplementary Fig. 5b, c), with the 3.0 Å structure, E603QCtAtm1$^{inw-opn}$, being almost indistinguishable from this state (RMSD 0.31 Å) and (Supplementary Fig. 11).

In contrast, while overall similar to CtAtm1$^{inw-opn}$ (RMSD 0.59 Å, Supplementary Fig. 11), the 2.9 Å structure, CtAtm1$^{inw-opn/cluster}$, displays a large plate-shaped additional feature with a volume of approximately 3500 Å³ in the cryo-EM density that is larger than the 370 and 740 Å³ volumes of GSH and GSSG (Supplementary Fig. 12a), respectively, somewhat peripheral to the electropositive cavity more toward the mitochondrial matrix, and an observation compatible with [2Fe-2S](GS)₄ cluster binding (Fig. 2a). Indeed, docking attempts reveal that the cluster can be accommodated within the observed cryo-EM density (Fig. 2b and Supplementary Fig. 12). Notably, the relative strength of the cryo-EM signal is clear from the assessment of the density at the

higher signal to noise ratios. At contour level 1.3, the protein is almost indistinguishable, but the additional feature remains clear (Supplementary Fig. 12). Notably, the cluster feature is also clearly reproduced in the relatively low-resolution cryo-EM data obtained from wild-type in the presence of AMP-PNP and the cluster (Supplementary Fig. 9).

While the distances are somewhat long, several electropositive conserved residues among eukaryotic Atm1 proteins appear to stabilize the cluster in this position, including R221, R225 of TM2, R289, R290, N293, N297 (less invariant) of TM4, and R402, E403 of TM6. This would suggest a weak charge-complementing interaction, based on the highly negative surface of the cluster. To further investigate the binding, we performed UV-visible spectrophotometry and inductively coupled plasma mass spectrometry (ICP-MS) measurements using protein–cluster complex forms purified following mixing (Fig. 2d, e). Indeed, wild-type exhibits two specific absorbance peaks in the 300–500 nm range that represent a fingerprint of proteins with bound [2Fe-2S] centers[36] (Fig. 2d). These features are missing for wild-type CtAtm1 that has not been supplemented with the cluster. The ICP-MS data also indicate iron is bound when the wild-type is mixed with the cluster, clearly above the threshold for wild-type alone (Fig. 2e). Furthermore, the interaction with the cluster is decreased by several multiple double and triple mutants of residues lining cluster in CtAtm1$^{inw-opn/cluster}$ (R285A/Q351A, R289/R402A, R289/R402A/Q351A) as the absorbance signal is lower (Fig. 2d), and ICP-MS demonstrates that this is associated with lower iron content than for the wild-type. Considering the location of the cluster in immediate connection to the inward-facing open cavity, we propose this state is poised for cargo uptake from the mitochondrial matrix.

### Cargo binding

How then do [2Fe-2S](GS)₄ clusters associate with Atm1 proteins for transport? Interestingly, the 3.2 Å map obtained from the above-mentioned E603Q sample in the presence of the cluster provided a distinct structure, CtAtm1$^{inw-opn-occl}$ (RMSD to CtAtm1$^{inw-opn}$ 1.83 Å). This configuration displays a considerably smaller distance between the NBDs (reducing the gap from around 42–27 Å (C$_\alpha$ distances between K480 and S574, of the Walker A and ABC signature motifs, respectively), but again with poorly resolved soluble domains, Supplementary Fig. 5), essentially forming an inward-open state with a less exposed, or partially closed/occluded, cavity (Fig. 3a). As such, the conformation is reminiscent of cargo-bound ScAtm1, AtAtm3 and NaAtm1 (PDB-IDs 4MYH[3,27]7N59[29] and 4MRS[3], RMSD 2.6, 2.0, and 2.0 Å, respectively) (Supplementary Fig. 13). Indeed, the presence of additional cryo-EM density is detected, coordinated by residues in immediate vicinity of N395, namely R285, R289 of TM4, N348, Q351 of TM5, and S399 of TM6 (Fig. 3b). These coincide with certain of the residues in the above-mentioned mutations (R285A/Q351A and R289A/R402A/Q351A) that display decreased cluster binding and lower iron content (Fig. 2d, e), further supportive of the notion that they partake in cluster cargo binding. However, the inner gate is substantially reconfigured in the shift from the inward-facing open states to the CtAtm1$^{inw-opn-occl}$ conformation, reorienting the N395 and F396 pairs, first pointing toward and away from the cavity, respectively, and then vice versa (Fig. 3c). The location of this putative cargo approximately overlaps with the single endogenous GSH molecule previously identified in the eukaryotic ScAtm1 (PDB-ID 4MYH[27]), GSSG in AtAtm3 (PDB-ID 7N59[29]), and with different bound cargo to the prokaryotic NaAtm1, such as GSSG (PDB-ID 4MRS[3]) (Fig. 3d, e).

As no GSH or GSSG was supplemented to the sample, we interpret this feature as a part of the cluster or even a complete cluster with one flexible half; however, conservatively modeled as two GSH molecules. Yet, no additional density is present as would be expected even for a partially occupied Fe−S center, considering its relatively high interaction with electrons compared to lighter elements. The CtAtm1$^{inw-opn-occl}$

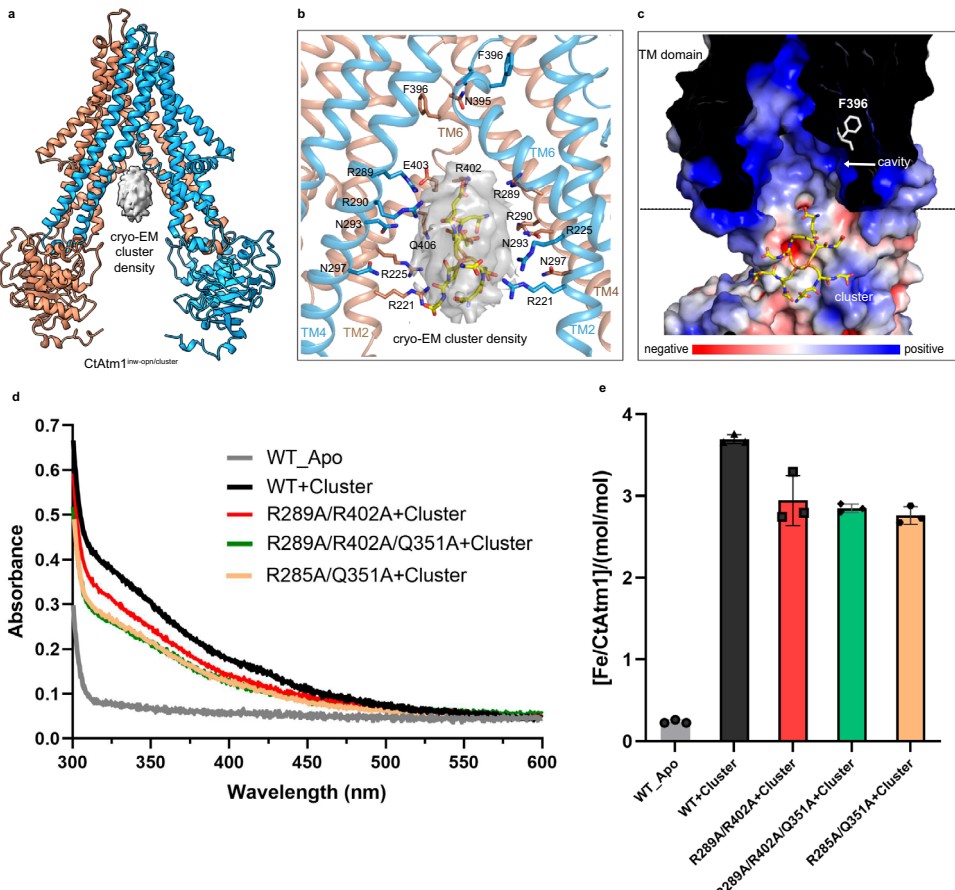

**Fig. 2 | The [2Fe-2S] cluster bound inward-open structure. a** The cluster bound inward-open structure, CtAtm1$^{inw-opn/cluster}$, with the cryo-EM density of the cluster (gray), and the protein colored as in Fig. 1d. **b** Close-view in the same orientation as in panel **b** of the cluster binding with the docking model shown in yellow. Residues within 8 Å of the bound cluster are shown as sticks. **c** The surface charge of the cavity and the associated cluster binding region in the transmembrane domain, cut as shown in panel **b**. The sidechain of F396 is shown as a stick. **d** UV-visible spectra of purified CtAtm1 variants with cluster and wild-type control (apo) without the cluster. **e** Iron binding to wild-type and mutant CtAtm1 forms treated with the cluster as deduced using ICP-MS (stoichiometries refer to CtAtm1 monomers). Data points represent the means of three independent measurements and error bars indicate standard deviation.

conformation also triggered us to revisit the initial cryo-EM data collected for wild-type CtAtm1 in the presence of ATP and without supplementation of cargo. Indeed, an equivalent state is present also in that data; however, the resolution of the information is lower (Supplementary Fig. 4). With the latter caveat, no indication of cargo is present in the cryo-EM density, consistent with cluster-induced uptake (Supplementary Fig. 14). Conversely, the two GSSG molecules bound to NaAtm1 are congruent with the notion that Atm1 proteins could accommodate and transport cargo species similar to [2Fe-2S](GS)$_4$ (Supplementary Fig. 1)[3]. Moreover, molecular docking of [2Fe-2S](GS)$_4$ guided by the cryo-EM density suggests that binding of the entire cluster is possible at this location in the CtAtm1$^{inw-opn-occl}$ conformation (Fig. 3f).

Aggregated, it thus appears as if the Atm1 family of proteins shares overall cargo-binding principles independent of whether heavy metal GSH/GSSG or (parts of) [2Fe-2S](GS)$_4$ clusters are transported. Nevertheless, the local chemistry of the interaction is rather different between the two subclasses, in agreement with bis-coordinated heavy metal ($M^{2+}$) forming a complex with glutathione with an overall 2− charge, while a tetra-coordinated [2Fe-2S] cluster complex carries an overall 6− charge, [2Fe-2S](GS)$_4^{6-}$. Consequently, in prokaryotic Atm1 proteins, a more electroneutral surrounding is present, including two methionines at the inner gate, NaAtm1 M317 and M320 of TM6, that directly links to the cargo, and even to the mercury atom of GS-Hg, in agreement with HSAB Pearson theory[37], postulating that many heavy

metals rely on soft ligand coordination by methionines and cysteines (Supplementary Fig. 3)[38]. In contrast, the situation is different in the eukaryotic proteins, with a highly electropositive environment and the methionine residues being replaced by F396 and S399 (the latter replaced by threonine in human ABCB7), respectively, in CtAtm1, seemingly incompatible with direct interaction with metal ions. Thus, the chemistry of the pocket and its overall charge appears to have a critical impact on the type of transported cargo among Atm1 proteins.

## Occlusion

Structures of occluded Atm1 are available for the eukaryotic ScAtm1 and AtAtm3, displaying an internal pocket without bound cargo (PDB-ID 7PSN[28] and 7N5A[29]). For the prokaryotic NaAtm1, the inward-facing occluded and occluded states (for example, PDB-ID 6PAM and 6PAO[30]), respectively, maintain a large internal cavity, although the transported species is not always caught within the pocket[30]. In addition, a closed post ATP hydrolysis NaAtm1 intermediate, following cargo release, has also been determined, without an internal cavity, thereby presumably ensuring directionality of the transport (PDB-ID 6VQT). The inward-open, inward-facing occluded, and the closed conformations have in common a substantial kink of TM6, and in agreement with this notion, this transmembrane segment is also bent in our above-mentioned structures of CtAtm1. In contrast, the shape of TM6 in occluded states of NaAtm1 is more straightened. To generate an occluded configuration, we again exploited the E603Q form, but

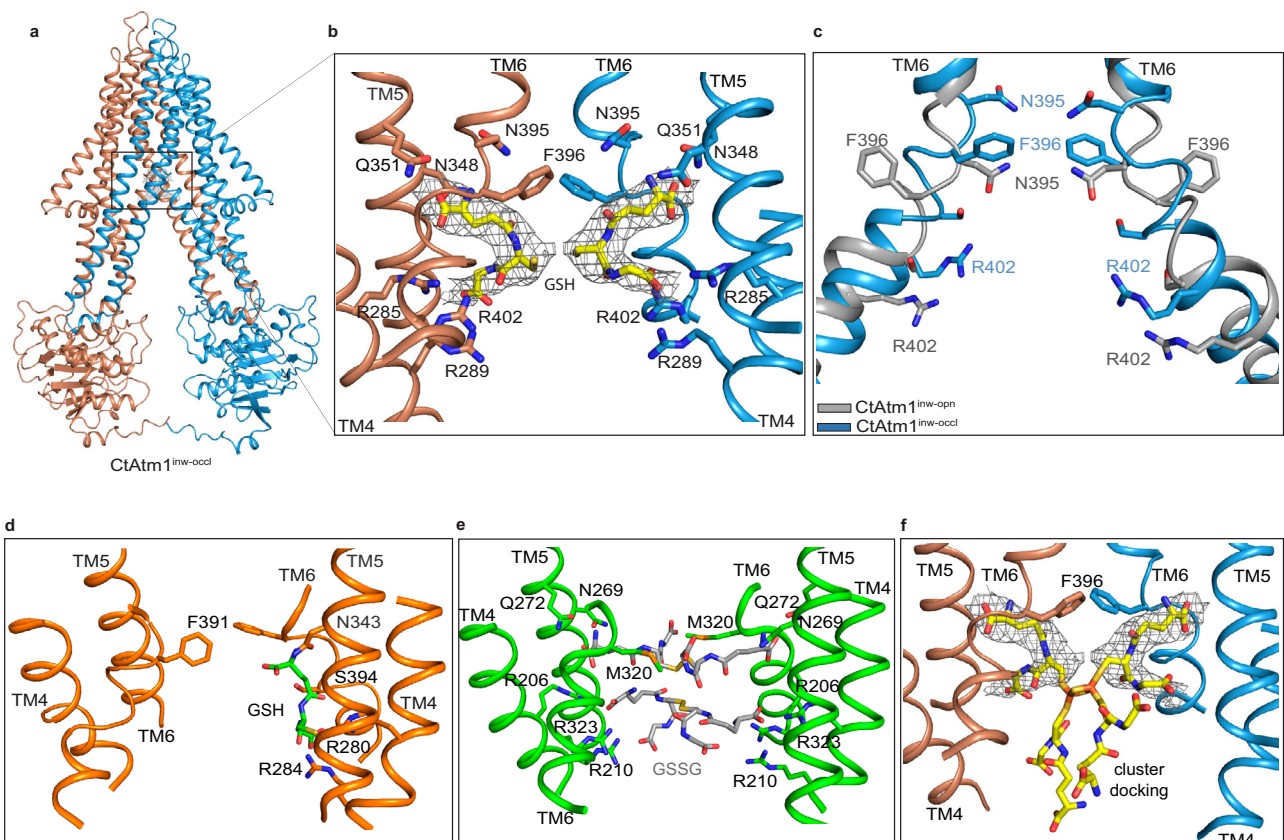

**Fig. 3 | The inward-facing partially occluded structure with bound cargo. a** The inward-facing partially occluded structure with bound cargo, CtAtm1[inw-opn-occl], with the cryo-EM density of bound cargo (gray) and the protein colored as in Fig. 1d. **b** Close-view of the cargo binding site. The cryo-EM density of the cargo is shown in gray and the modeled glutathione (GSH) molecules as yellow sticks. Asn395 and F396 and residues involved in cargo binding are also shown as sticks. **c** Comparison of the conformation of transmembrane helix TM6 between the inward-open (CtAtm1[inw-opn], gray) and the CtAtm1[inw-opn-occl] (blue) states, demonstrating a considerable rearrangement of F396. **d** Identical view as in panel **b** of ScAtm1 with bound GSH (PDB-ID 4MYH). **e** Identical view as in panel **b** of NaAtm1 with bound oxidized GSH (GSSG) (PDB-ID 4MRS). **f** Identical view as in panel **b** with the glutathione-bound [2Fe-2S] cluster, [2Fe-2S](GS4), partially fitting in the cargo density.

this time we supplemented the sample with ATP and cargo, thereby resembling conditions previously exploited to catch such a state of NaAtm1[30]. First, we attempted to include the [2Fe-2S](GS)4 cluster, but these efforts were fruitless, as a result of protein precipitation. Then, the cluster in the sample was replaced with GSSG, which permitted us to determine a structure at an average resolution of 2.8 Å, CtAtm1[occl]. In contrast to previous forms, this sample demonstrates well-defined NBDs in which two ATP molecules are readily identified in the cryo-EM density in proximity to Y450, K480, S481, and Q603 (Fig. 4a and Supplementary Figs. 5 and 15–17).

As anticipated, this structure is alike the equivalent state of ScAtm1 and AtAtm3 (RMSD 1.6 and 1.4 Å, Supplementary Fig. 18). Unexpectedly, CtAtm1[occl] however demonstrates mixed properties, showing the highest resemblance to the fully occluded NaAtm1 state (PDB-ID 6PAO, RMSD 1.6 Å), and yet also a high similarity to the closed post ATP hydrolysis intermediate (PDB-ID 6VQT, RMSD 1.9 Å) (Supplementary Fig. 18). In contrast, CtAtm1[occl] is more dissimilar to the inward-facing occluded NaAtm1 conformation (RMSD 3.8 Å). Indeed, surface analysis reveals the TM-domain to remain inaccessible from the intermembrane space of mitochondria, but now also from the matrix side. Distinct from the closed (post ATP hydrolysis) NaAtm1 structure, a pocket is present, although relatively small, which however is empty as also observed in the occluded NaAtm1 states and for ScAtm1 and AtAtm3 (Fig. 4b). Yet, TM6 remains kinked in the structure, in contrast to occluded states of NaAtm1 (Fig. 4c, d). In this light, we conclude the structure likely represents an early fully

occluded state, triggered by the supplementation of cargo and/or nucleotide. We note it has previously been proposed that the shape of TM6 orchestrates occlusion, providing uptake/space/release and eliminating the cavity in the straightened and kinked arrangements, respectively[29,30]. This however appears only partly correct in the eukaryotic proteins, as the pocket region is detected in CtAtm1[occl] with TM6 persisting kinked, while it is straightened in the outward-open structure of AtAtm3 (Fig. 4c). The observation perhaps illustrates minor mechanistic differences between the Atm1 subclasses, and resonates with the higher molecular weight of the transported cargo of eukaryotic members, presumably precluding the more closed configuration achieved in the prokaryotic NaAtm1.

The overall location of the CtAtm1[occl] pocket across the matrix-membrane interface between TM2, TM3 and TM 6 has essentially not been altered vis-à-vis the matrix-exposed cavity present in the inward-facing conformations, and the highly electropositive surface charge is also maintained (Fig. 4b). It is lined by residues including R221, R225, K228, R402, and H240 that have a maintained chemical character in eukaryotic but not prokaryotic Atm1 proteins (Supplementary Fig. 3). However, interestingly, in the cargo-binding region next to F396, the pocket is divided by residue pairs Y232, V400, and E403 aligning directly each side of the pocket with the two pathways separated by the TM6 linking pairs that are present in our other CtAtm1 structures (Fig. 1g and Supplementary Fig. 7). Continued passage through the TM-domain is however restricted by the now enlarged inner gate, consisting of H240, F396, and S399 (Fig. 4b). Nonetheless, the pocket is

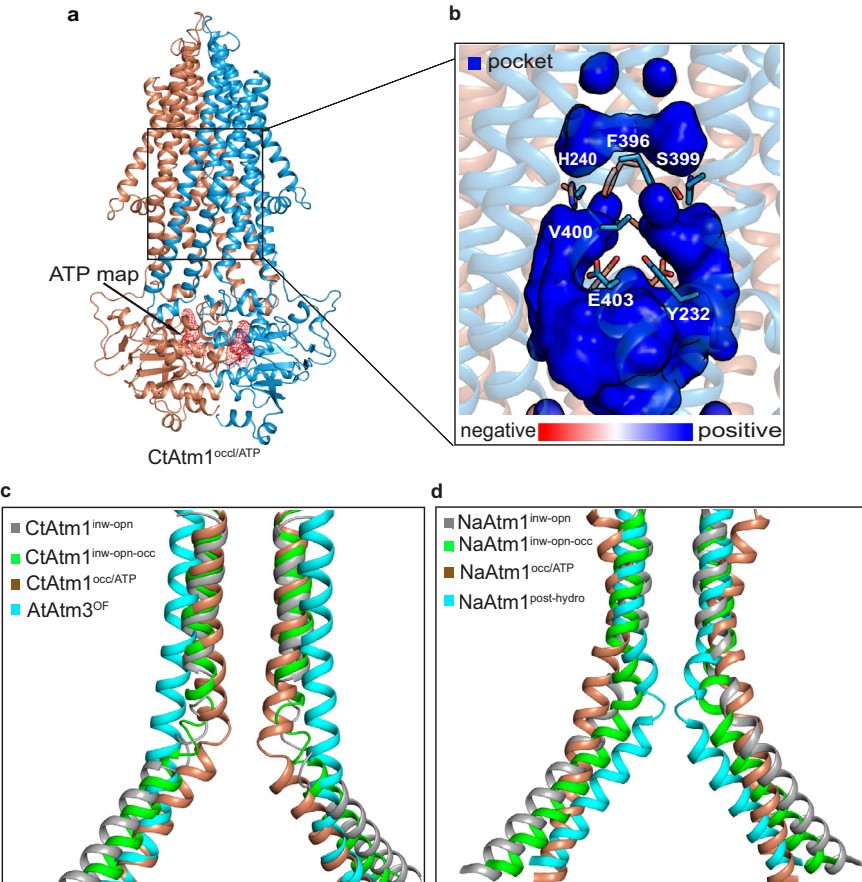

**Fig. 4 | The occluded structure with bound ATP. a** The occluded structure with bound ATP, CtAtm1$^{occl/ATP}$, with the supporting cryo-EM density for ATP (red) and the protein colored as in Fig. 1d. **b** Close-view of the occluded highly electropositive pocket. Residues Y232, E403, and V400 divide the pocket next to the inner gate. The interactions between H240, F396, and S399 separate the pocket from the pathway on the other side of F396. **c** Comparison of the conformation of transmembrane helix TM6 between the inward-open (CtAtm1$^{inw-opn}$, gray), the inward-facing partially occluded (CtAtm1$^{inw-opn-occl}$, green), the occluded (CtAtm1$^{occl}$, brown), and AtAtm3 outward-facing (AtAtm3$^{OF}$, cyan) states, demonstrating a considerable rearrangement of F396. **d** Identical view as in panel **c** of the equivalent states of the prokaryotic Atm1 member NaAtm1: inward-open (PDB-ID 6VQU, gray), inward-facing occluded (PDB-ID 4MRS, green), and occluded (PDB-ID 6PAO, brown). In addition, the closed post-hydrolysis state of NaAtm1 is shown (PDB-ID 6VQT, cyan).

still able to accommodate an entire [2Fe-2S](GS)$_4$, as supported by complementary molecular docking efforts (Supplementary Fig. 19). This suggests the trapped occluded conformation likely represents an early occluded state, with the pathway split due to the kink of TM6. Later, straightening of TM6 (Fig. 4c) likely merges and extends the cavity to the outward-facing (intracellular) side, permitting export.

## Release

Hitherto, a single structure of an outward-facing conformation of an Atm1 protein, AtAtm3, has been elucidated, hence limiting the understanding of how cargo release takes place in these transporters, also because this structure was lacking cargo[29]. We have also been unable to recover such an intermediate. Based on the AtAtm3 structure with a considerably wide pathway to the cellular inside, we note that the plausible export path, from the occluded cargo-binding region next to the inner-gate region, directly through the TM-domain, is lined by the TM1 and TM6 helix pair, which have become straightened, and several other helices (Fig. 4c and Supplementary Fig. 7). This is also supported by the far penetrating parallel pockets in the structures, beyond the obstructed inner gate (Figs. 1g, 2c, and 4b), which end in proximity to a highly conserved methionine pair, M280, and M284 in NaAtm1. It is possible that these stimulate the export process in prokaryotic Atm1 proteins, perhaps assisting in forming an ion-

binding site that directly interacts with the cargo as observed at the inner gate. However, the situation is different in the eukaryotic members, which only have maintained one of these residues, M363 in CtAtm1, equivalent to M284 of NaAtm1 (Supplementary Fig. 3). Furthermore, the equivalent of M363 is not exposed to the intracellular environment and is hence not likely participating in release.

## Transport mechanism

Previous studies have established that Atm1 ABC transporters function according to the switch model. Briefly, the cargo and ATP are able to associate with the protein in an inward-facing conformation, interacting with residues at the cargo-binding site and NBD, respectively. Occlusion of the transported species, bound in an internal cavity, and subsequent cargo release through an outward-facing state, is triggered by the concomitant presence of nucleotide and cargo. This is followed by a major rearrangement of the protein, as achieved by a closed configuration without an internal pocket, at least in the prokaryotic members. Finally, ATP hydrolysis reverts the protein back to the inward-open conformation. The global conformational changes that permit this transport activity seem conserved among ABC transporters[3]. The cross-membrane activity of all Atm1 proteins appears to be facilitated by GSH, but while prokaryotic members are transporters of heavy metals as complexes with GSH, fungal and

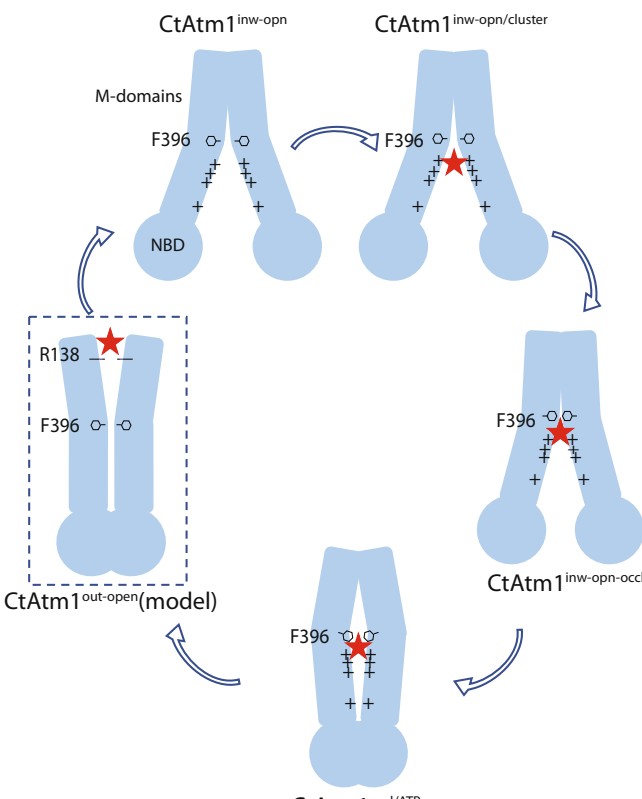

**Fig. 5 | Mechanism of cluster transport by Atm1.** We propose the inward-open configuration represents the resting state of eukaryotic Atm1 proteins, in which the protein exposes a highly electropositive cleft to the inside of the mitochondrial matrix, as detected in the CtAtm1inw-opn (and E603QCtAtm1inw-opn) structure. This state permits negatively charged glutathione-bound [2Fe-2S] clusters (red star) to associate in the vicinity of the cleft, as observed in our CtAtm1inw-open/cluster structure. Cargo and likely nucleotide binding to the positively charged cavity in the transmembrane domain and the nucleotide-binding domain, respectively, trigger conformational changes that result in reorganization of the F396 inner gate, partial occlusion (as in the CtAtm1inw-opn-occl structure) and then complete occlusion (as in the CtAtm1occl/ATP structure), with local chemistry of the cargo-binding pocket that is distinct to prokaryotic Atm1 proteins. We anticipate the [2Fe-2S] cluster or (an) intermediate(s) thereof is enclosed in the occluded state. Release to the intracellular side is likely coupled to further conformational adaptation, straightening of TM6 to permit passage across the inner gate, and opening the transport pathway. Once the cargo is liberated from the protein, ATP hydrolysis takes place, permitting a new transport cycle.

human proteins of the same family have been implicated in the mobilization of [2Fe-2S] clusters[18,19,31,34].

As revealed by the inward-facing open CtAtm1inw-opn/cluster structure with bound [2Fe-2S](GS)$_4$, we propose that the overall negatively charged cluster weakly associates through electrostatic complementation with the linking helical stretches between the TM-domains and the NBDs, which expose positively charged residues (Fig. 5). This must be considered strong evidence that eukaryotic Atm1 proteins transport, or at least, accept delivery from glutathione-complexed [2Fe-2S] clusters, although it is possible that additional cargo types may be transported, as also indicated by the observed stimulation by GSH and GSSG (Fig. 1c). Considering the overall similarity to the prokaryotic counterparts, we, however, anticipate that cargo recognition and transport mechanism are preserved overall.

Next, the cargo is recruited to conserved motifs of the binding pocket, coupled to initiation of occlusion and perhaps nucleotide binding, as observed in the partially occluded CtAtm1inw-occl

structure. Our data are inconclusive regarding the nature of the bound cargo because only parts of the cluster are detected, although flexibility may partly explain this observation. However, considering that two GSSG molecules, together having approximately the same size and with chemical similarity to [2Fe-2S](GS)$_4$, have been identified at this location for NaAtm1, we cannot exclude that eukaryotic Atm1 proteins also have the capacity to bind the intact [2Fe-2S] cluster. Alternatively, following the initial delivery of [2Fe-2S](GS)$_4$, cluster intermediates such as [2Fe-2S](GS)$_2$ are transported, which may be stabilized in the cargo site from hydrolytic degradation, a molecular species that would be congruent with the approximate molecular weight of the cargo of prokaryotic Atm1 members. Recruitment and binding of the highly negatively charged cluster is presumably guided by the highly electropositive characteristics of the cavity, more highly charged than the initial location of the cluster, toward the matrix (in the CtAtm1inw-opn/cluster structure). In this state, further passage to the mitochondrial intermembrane space is limited by the inner-gate residues, including F396 and H240 (or the equivalent as observed in ScAtm1 and AtAtm3), separating the eukaryotic and prokaryotic subclasses, as the inner gate of the latter includes a methionine that is directly involved in metal binding. Furthermore, the pocket of prokaryotic Atm1 proteins seems significantly less electropositive, in agreement with the less charged cargo.

Subsequently, full occlusion is achieved, a conformation in which the cargo is inaccessible for the surrounding environment (Fig. 5). In this structure, CtAtm1occl, we were unable to detect internalized cargo, as has also been the case for all occluded structures Atm1 proteins. However, a considerable pocket remains, although the part of it that is in direct contact with the inner gate has been divided into two portions (Fig. 4b). We propose that intact clusters or intermediates thereof persist at the matrix side of the inner gate, but become poised for subsequent export through dislocation to interact with H240 and F396. Intact cluster binding within the larger part of the cavity, more toward the matrix, is possible as shown by our molecular docking efforts. Likewise, it cannot be ruled out that the cargo, in the transport mechanism, is occluded closer toward the cytoplasm, as the inner-gate restricted pockets that line TM6 protrude in that direction along the expected export pathway (Fig. 4b).

Eventually, cargo release occurs, associated with outward-facing configurations. Displacement of the cargo from the pocket is likely stimulated by reorganization of the inner gate, as detected from the CtAtm1occl to the CtAtm1inw-opn state (Fig. 3c). We again note the passage across the membrane of the entire cluster may be disfavored by the apparent division, on either side of N395 or F396, of the export pathway from the inner gate throughout the generated CtAtm1 structures (Figs. 1g and 4b). However, considering that the export pathway expands significantly as observed in the outward-open AtAtm3 structure, the export of the intact cluster cannot be excluded. The arrangement, with loops exposed to the cytoplasm, may also be compatible with re-ligation or even docking of cytosolic acceptor proteins to provide stabilization of the exported cluster intermediate, a process that then likely would be independent of M363, and rather involve charged residues such as R138 and Q385, which structurally overlay K165 and Q428 in AtAtm3.

## Discussion

The structural and functional efforts presented here provide insights into the molecular mechanism of Atm1 proteins, and in particular on [2Fe-2S] transport permitted by eukaryotic members, pinpointing shared molecular principles and differences between prokaryotic and eukaryotic Atm1 transport. As with other eukaryotic members, CtAtm1 accepts a delivery from such metal centers for transport across membranes, and strong indications of a

glutathione-complexed [2Fe-2S] cluster are identified in an inward-facing open structure, in immediate connection to the cargo-binding pocket. The cavity demonstrates unique electrostatic properties, and different local chemistry at the limiting inner gate, distinguishing prokaryotic and eukaryotic Atm1 members. Interaction with the cargo within the cavity triggers occlusion, in which the cavity is internalized close to the inner gate, presumably compatible with transport of an intact $[2Fe-2S](GS_4)$ cluster or an intermediate thereof. It is expected that release toward the cytoplasm occurs via a wide outward-open pathway. While these findings provide critical new light on molecular determinants of eukaryotic Atm1 proteins, additional structural and functional characterization is essential to fully decipher the Atm1 transport mechanism.

## Methods

### Cloning, overproduction, and purification

The full-length CtAtm1 (Uniprot G0SBE6) was amplified from the genome of *C. thermophilum* (DSM1495) with primers CtATM1-F/CtATM1-R (Supplementary Table 2). The PCR fragments were digested using NcoI and SacI, and the digestion product ligated into the vector pET-52b, with the thrombin cleavage site replaced by a TEV cleavage site, yielding pET-52b-CtATM1-TEV-His$_8$. The integrity of the construct was confirmed by sequencing. This was then employed as a template for the generation of all mutant forms, which were obtained through overlap PCR with individual primers pairs (Supplementary Table 2). The plasmid was transformed into the C43 (DE3) *E. coli* strain. Single colonies were inoculated in 10 mL LB medium supplied with 100 μg/mL ampicillin at 37 °C with shaking at 180 rpm for 16 h. The preculture was then transferred into 500 mL TB culture supplied with 2 mM $MgCl_2$, 100 μg/mL ampicillin, 40 μg/mL ferrous ammonium citrate, and incubated at 37 °C with shaking at 180 rpm. Protein expression was induced with 0.8 mM isopropyl β-D-1-thiogalactopyranoside at 20 °C for 20 h, when $OD_{600nm} = 1.0$. The cells were harvested through centrifugation at 6000 × g for 30 min. The cell pellets were then washed with PBS buffer and resuspended with lysis buffer (10 mM Tris pH = 8.0, 250 mM NaCl, 5% (v/v) glycerol, 2 mM 2-mercaptoethanol (BME), 2 mM $MgCl_2$, 2 mM EDTA). The cells were disrupted by sonication for 30 min. Unbroken cells and cell debris were removed by centrifugation at 12,000 × g for 20 min, and the membranes were collected through ultracentrifugation at 160,000 × g for 1 h. The pelleted membranes were solubilized at 4 °C for 2 h in the presence of buffer A (25 mM Tris pH = 8.0, 500 mM NaCl, 10% (v/v) glycerol, 2 mM BME) supplemented with 1% (w/v) DDM. The insoluble material was removed by ultracentrifugation at 190,000 × g for 30 min. The supernatant was loaded on a prepacked 5 mL Histrap column (Cytiva), equilibrated with 30 mL buffer A containing 0.05% (w/v) DDM. Contaminations and unbound protein were removed through the washing of the column with 10 column volumes of buffer A, supplemented with 0.05% (w/v) DDM and 50 mM imidazole. The target protein was eluted with buffer A containing 0.05% (w/v) DDM and 300 mM imidazole. Fractions containing CtAtm1 were pooled and concentrated using 100 kD MWCO concentrators (Sigma). The sample was further polished using size-exclusion chromatography (SEC) using a Superose 6 column (Cytiva). The SEC profiles of all studied CtAtm1 forms are shown in Supplementary Fig. 20. The peak fractions with CtAtm1 were pooled and concentrated to 10 mg/mL, and stored at −80 °C until used for downstream applications. The protein purity was verified using SDS-PAGE. Based on these procedures, we could typically isolate 5 mg purified protein from 1 L of cultured cells.

### Nanodiscs reconstitution

MSP1D1 was purified and used for the nanodisc reconstitution as previously reported[33,39]. Wild-type CtAtm1 was reconstituted into nanodiscs using DMPC lipid with a molar CtAtm1:MSP1D1:DMPC ratio of 1:2.5:100, conducted at 25 °C for 3 h. The E603Q form was reconstituted into nanodiscs using POPC lipid with a molar CtAtm1(E603Q):MSP1D1:POPC ratio of 1:1.5:100, conducted at 4 °C for 16 h. In all, 200 mg bio-beads (Bio-Rad) were employed to remove detergent for each sample. The reconstituted nanodiscs sample was further purified using immobilized metal ion affinity and then SEC to remove nanodiscs without target protein, the latter in the presence of 20 mM Tris pH = 7.5, 100 mM NaCl. The final CtAtm1 nanodiscs in the peak were pooled and concentrated to around 1 mg/mL for the cryo-EM study.

### Cryo-EM sample preparation, data collection, and image processing

Wild-type CtAtm1 in DMPC nanodiscs was frozen at a concentration of about 1.0 mg/mL. Quantifoil 1.2/1.3 holy carbon grids were glow-discharged using a Leica Coater ACE 200 for 30 s with 10 mA current. The grids were prepared using a Vitrobot Mark IV operated at 100% humidity and 4 °C. In all, 3 μL of purified protein was applied to each grid, incubated for 5 s, blotted for 3 s, and then plunge frozen into liquid ethane. Frozen grids were stored in liquid nitrogen until data collection. For the CtAtm1$^{inw-opn}$ conformation, wild-type CtAtm1 in POCP nanodiscs was incubated with 1 mM ATP and 2 mM $MgCl_2$ for 30 min on ice prior to freezing. For the CtAtm1$^{inw-opn/cluster}$ and CtAtm1$^{inw-occl}$ states, wild-type or E603QCtAtm1 in POCP nanodiscs was incubated with 1 mM of $[2Fe-2S](GS)_4$, 1 mM AMP-PNP and 1 mM $MgCl_2$ for 5 min at 40 °C (water bath) prior to freezing. For the CtAtm1$^{occl}$ conformation, the same E603Q form in POCP nanodiscs was incubated with 1 mM GSSG, 1 mM ATP, and 1 mM $MgCl_2$ for 5 min at 40 °C prior to freezing. The cryo-EM datasets were collected on Titan Krios electron microscopes (FEI) operated at 300 kV with a Falcon3 detector in counting mode. The pixel size was set to 0.832 Å and the total dose was 40 e/Å$^2$ in 40 frames. Processing details using cryosparc are shown in Supplementary Figs. 4, 9, 10, and 16 and Supplementary Table 1. For the CtAtm1$^{inw-opn}$ data set, cryosparc 3D variability analysis provided information on the structural heterogeneity suggesting differences occur in the NBDs (Supplementary Fig. 5). The local resolution of the different generated cryo-EM maps has been estimated (Supplementary Fig. 21).

### Model building

The template employed for the model building was initially generated using the SWISS-MODEL server[40] with the CtAtm1 sequence and the crystal structure of ScAtm1 (PDB-ID: 4MYC) as entries. The generated Swiss model was fitted using a rigid body into the cryo-EM density map in UCSF chimera[41]. The model was manually adjusted in Coot[42] and refined using phenix_real_space_rfine[43]. MolProbity implemented in Phenix was used for model validation[44].

### Molecular docking

The initial coordinates of $[2Fe-2S](GS)_4$ cluster were obtained from the Chemical Entities of Biological Interest (ChEBI) database (ChEBI ID 167627) and optimized in PyMol. The cluster model was manually docked to the cryo-EM structures of CtAtm1, taking into account the available cryo-EM density, or the cavity in the proposed binding site. The models were finally adjusted using the PyMol sculpting module.

### ATPase activity assay

The CtAtm1 sample in DDM micelles or nanodiscs sample for the ATPase activity assay was purified as described above. The ATPase activity was measured using the ATPase/GTPase Activity Assay Kit (Sigma-Aldrich, product number MAK113). The exploited protein concentration for the assays was 0.05 mg/mL (final concentration with 50 μL reaction volume). The reaction buffers for the samples were the SEC buffer with 5 mM $MgCl_2$. Reactions were through supplementation of ATP (1 mM, final concentration). The assay was

carried out at 37 °C for 10 min, and 40 μL reaction sample was prepared and stopped through the addition of the 160 μL reagent from the assay kit. The mixture was then incubated at 18 °C for 15 min and the activity was measured by monitoring the absorbance at 620 nm using a 96-well plate setup (Fig. 1c). Other measurements were performed using a final protein concentration of 0.01 mg/mL using an unconcentrated sample from the SEC peak, and 20 μL reaction sample was mixed 80 μL reagent and the absorbance at 620 nm was recorded with 384-well plates (Fig. 2d). Statistical analysis was performed using GraphPad Prism.

## Cluster-binding assay

The cluster was supplemented with purified protein at a molar ratio of 5:1 and incubated at 18 °C in the dark. The complexed sample was then purified using a 5 mL Hitrap desalting column (Cytiva) with buffer C (50 mM HEPES, pH = 8, 150 mM NaCl, 10% (v/v) glycerol, 5 mM BME, 2 mM $MgCl_2$ and 0.03% (w/v) DDM). The protein fractions in the eluted sample were pooled and concentrated with 100 kD MWCO concentrators (Sigma) to 7.5 mg/mL. UV-visible spectra were recorded on a Cary 60 UV-vis spectrophotometer. Statistical analysis was performed using GraphPad Prism.

## ICP-MS

CtAtm1 samples for which cluster was supplemented to purified protein as described above binding were purified using a 5 mL Hitrap desalting column (Cytiva) with buffer C (50 mM HEPES, pH = 8, 150 mM NaCl, 10% (v) glycerol, 5 mM BME, 2 mM $MgCl_2$ and 0.03% (w/v) DDM). Iron content was quantified by ICP-MS (Agilent 7900) after digesting the samples in 50% $HNO_3$ (v/v) overnight at 85 °C and subsequently diluting each sample to a final 1% $HNO_3$ (v/v) using ultrapure $H_2O$. Final iron concentrations were calculated accounting for dilution, prior to the determination of experimental Fe-to-protein ratios ($n = 3$, average of three technical replicates).

## Reporting summary

Further information on research design is available in the Nature Research Reporting Summary linked to this article.

## Data availability

The structural coordinates and EM data have been deposited in the Protein Data Bank and in the Electron Microscopy Data Bank with the following accession numbers: CtAtm1$^{inw-opn}$ (PDB-ID 7PQX [https://doi.org/10.2210/pdb7pqx/pdb], EMD-13606), E603QCtAtm1$^{inw-opn}$ (PDB-ID 7PSD [https://doi.org/10.2210/pdb7psd/pdb], EMD-13612), CtAtm1$^{inw-opn/cluster}$ (PDB-ID 7PRO [https://doi.org/10.2210/pdb7pro/pdb], EMD-13609), CtAtm1$^{inw-opn-occl}$ (PDB-ID 7PRU [https://doi.org/10.2210/pdb7pru/pdb], EMD-13610), and CtAtm1$^{occl/ATP}$ (PDB-ID 7PR1 [https://doi.org/10.2210/pdb7pri/pdb], EMD-13607). All data and materials supporting the findings in the manuscript are available from the corresponding author upon reasonable request. Source Data are provided with this paper.

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

## Acknowledgements

We would like to thank Tillmann Hanns Pape at the Danish Cryo-EM Facility at CFIM, University of Copenhagen, for assistance with the Cryo-EM data collection. CFIM is supported by Novo-Nordisk Foundation grant id NNF14CC0001. We thank Julian Conrad and Karin Wallden for Cryo-EM sample screening at the Cryo-EM Swedish National Facility funded by the Knut and Alice Wallenberg, Family Erling Persson and Kempe Foundations, SciLifeLab, Stockholm University, and Umeå University. We thank Susanna Törnroth-Horsefield for providing access to the UV-vis spectrophotometer and Tamim Al-Jubair for assistance with the measurements, and Viktoria Bågenholm for help with the sample delivery, Christina Grønberg for LMNG sample preparation, and Maria Gourdon for initial crystallization experiments. We acknowledge access to the computational resources from the Danish National Super-computer for Life Sciences (Computerome). K.W. is supported by the Lundbeck Foundation. P.G. is supported by the following Foundations: Lundbeck (R313-2019-774 and R346-2020-2019), Knut and Alice Wallenberg (2015.0131 and 2020.0194), Carlsberg (2013_01_0641, CF15-0542, and CF21-0647), Novo-Nordisk (NNF13OC0007471), Brødrene Hartmann (A29519), Agnes og Poul Friis, Augustinus (16-1992), Crafoord (20170818, 20180652, 20200739, and 20220905) as well as The Per-Eric and Ulla Schyberg (38267). Funding is also obtained from The Independent Research Fund Denmark (9039-00273), the Swedish Research Council (2016-04474), and through a Michaelsen scholarship. Y.W. acknowledges the financial support from the Fundamental Research Funds for the Central Universities, China, and access to computational resources from the Information Technology Center and State Key Lab of CAD&CG, ZheJiang University. G.M. is supported by the National Institute of General Medical Sciences, National Institutes of Health (R35GM128704). The funders had no role in study design, data collection and analysis, decision to publish, or preparation of the manuscript.

## Author contributions

P.L. initiated the project and performed cloning, overproduction, purification, and nanodisc reconstitution for the cryo-EM efforts, the ATPase assay, the UV-visible spectroscopy, and the ICP-MS samples preparation. Cryo-EM sample grids preparation, data collection, and processing were primarily conducted by K.W. The cryo-EM data set were collected at CFIM University of Copenhagen. P.L. and K.W. built and refined the structures. Y.W. conducted the molecular docking analysis. A.L.H. and J.A.C. provided the cluster. R.L.E.V. and G.M. conducted the ICP-MS measurements. K.L.-P. facilitated the ATPase assay and the UV-visible spectroscopy experiments. P.L. prepared the figures except for Fig. 5 and Supplementary Figs. 4, 9, 10, 16, and 21 that were generated by K.W. P.L., K.W., J.A.C., and P.G. contributed to the identification of scientific problems and experimental planning. All authors conducted data analysis and interpretation. P.L., J.A.C., and P.G. wrote the first draft. All authors commented on the manuscript. K.W. and P.G. supervised the project.

## Funding

## Competing interests

The authors declare no competing interests.
