## [Peer Review File · Nature Communications]

REVIEWER COMMENTS

Reviewer #1 (Remarks to the Author):

The manuscript presents new structures of an Atm1 homologue, obtained by cryoEM. This is an excellent technical achievement, however the results are very similar to those published in 2014 by the groups of Lill (yeast Atm1) and Rees (bacterial Atm1 homologue). The discussion of important residues and comparison of the new structures with the previous structures is very informative.

The title maybe misleading? The words "Structures of Fe-S cluster export" suggests movement, but the data mainly consist of Cryo-EM structures, which are static. Moreover, there is insufficient resolution to be certain about the Fe-S cluster in one structure which supposedly has it, whereas the other structures do not have Fe-S bound (but GSSG).

Abstract "these clusters are primarily biosynthesized in mitochondria and then exported to the cytoplasm using ABCB7-type proteins" – the evidence in the literature for this is not strong, by all means, but this is not reflected in the wording here. Note, in the Introduction, the authors state "Even the type of cargo for the eukaryotic homologs has not been clearly elucidated."

Key papers on this topic reporting export of GSSG and persulfide are ignored. For example, Schaedler et al 2014 JBC, Pandey et al 2019 JBC. These papers should be summarized and cited in the Introduction, to balance self-citations 11 – 13 and 21.

Fig 1b. Are the E>Q mutant proteins expressed correctly, e.g. at comparable levels with wild type protein and not affected in their folding? The ATPase activity of the mutant proteins without cargo is unusually low. Every protein is different in this respect, and 'classical' amino acid substitutions that work well for one ABC transporter, may not work for an orthologue from another species. Please add information on the typical protein yield per liter bacterial culture and show Coomassie-stained gels of the purified proteins. What are ATPase activities with cargo, e.g. GSSG? No data are given for this.

The Fe-S/GSH cargo is deduced from modelling and docking studies. Evidence for binding of the cargo, and that this is physiological, should be provided. For example by mutagenesis of the amino acids that are modelled to interact with the 'disc-like structure'. A UV-vis spectrum of the purified protein binds Fe-S/GSH should be shown.

Reviewer #2 (Remarks to the Author):

This was an excellent and exciting paper overall; the ability of the authors to obtain a high-resolution cryo-EM structure of an Atm1, with cargo bound, is remarkable and highly significant. The authors used very creative means of preventing the translocation of cargo (inactive mutant protein and derivative of ATP that cannot be hydrolyzed) in obtaining their structures. Nevertheless, I have a few concerns. My major concern is the need to clarify the degree to which the authors are certain that the density observed upon adding the cargo is in fact an $[2Fe_2S](GSH)_4$ cluster. They state unambiguously in the abstract and discussion section that this has been "demonstrated". However, they seem to be more cautious in the description in the Results section and I wasn't able to discern the strength of their evidence. There is a long history to this. On page 2, the authors state that the $[2Fe_2S](GSH)_4$ cluster "is of interest because such complexes are stable in aqueous media thereby permitting $[2Fe_2S]$ cluster to be mobilized in the cell without a protein scaffold (Ref 12 – 15)". However, many researchers in this field are skeptical that $[2Fe_2S](GSH)_4$ clusters can exist stable in aqueous solution. All of the references supporting this statement are from the same lab, as no other lab has confirmed this. I am agnostic on this issue, but am certain that all of the doubters would be silenced immediately if the authors would publish two strong pieces of evidence including: a) a full high-resolution ESI-MS spectrum of the cluster after being incubated in aqueous buffer for a few days, showing the predicted masses including the expected isotopomers in the expected intensity ratios; and b) Mossbauer spectra of the same, showing a clean quadrupole doublet with parameters of a $[2Fe_2S]^{2+}$ cluster and with no cluster degradation after incubation. With respect to the current data, I couldn't tell whether the authors were able to establish that the cargo definitely contains iron. Is the size and shape of

the cargo similar to what would be expected if it were due to this cluster? This issue should take "center stage" in the current paper.

The only other concern was that the history of the Atm1 story was not well developed in the introduction. The recent papers from the Pain/Dancis lab on this issue (Pandey et al) were not mentioned but should be. Relevant work by Janneke Balk on glutathione trisulfide should be mentioned as well. Finally, the title of the article should read Atm1 not Amt1. They should also mention in the title and abstract the source of Atm1 used in their study.

Reviewer #3 (Remarks to the Author):

In this manuscript, Li et al. employ cryo-EM to investigate the mechanism of [2Fe-2S] export by eukaryotic ABC transporter Atm1. As their model system, the authors use nanodisc-reconstituted CtAtm1 from thermophilic fungus. Utilizing ATP or AMP-PNP trapping, the authors report a series of structures, providing important insights into the transport mechanism and selectivity of the cargo. I have no doubt that this manuscript would be of high interest to many researchers in ABC transporter field, however, I still have some concerns that should be addressed.

1. The authors obtained the structure of wild-type CtAtm1_{inw-opn} in the presence of ATP, however, in the methods section there is no information on how exactly this particular experiment was performed: what concentration of ATP and Mg²⁺ was used and at which temperature and for how long the sample was incubated prior to grid preparation. The authors should add the missing information.
2. On page 5, the authors write: "... the resolution of the NBDs is somewhat lower, and hence we are unable to confirm the absence of nucleotide binding". Lower resolution at the NBDs is not surprising due to their flexibility, but looking at the processing scheme in Supplementary Fig. 4, it does not seem that the authors tried to improve the densities at the NBDs, for example, by performing local refinement with or without symmetry expansion. I would recommend to do that, because I am quite surprised that the authors expect the absence of nucleotides even though they added ATP. I would expect that the ATP should have been hydrolyzed, but ADP is still bound in the structure - in such a case, I would also say that the authors statement that it is "a pre-translocation arrangement" is debatable, but it is really hard to judge this without having any details on how this experiment was actually performed, as already mentioned in my first comment.
3. In addition, while the authors report only one final structure from this dataset, Supplementary Fig. 4 displays several different conformations with variable degree of NBD separation (best visible in 2D classes). Did the authors look into this and tried to refine the different conformations separately instead of just going for a single highest resolution structure?
4. It is also not clear to me, why for trying to capture the structure with the cargo, the authors utilize both the EQ mutant and AMP-PNP. Normally, one would go for either EQ mutant with ATP or wild-type with AMP-PNP, but the authors chose the option with both modifications - is there any specific reason for that? Since the authors didn't obtain any structures with dimerized NBDs from this dataset, AMP-PNP trapping seems to simply have not worked and the same structures would have been probably obtained without addition of any nucleotides. Did the authors try to obtain such data? Even though the trapping didn't work, the structures are still valuable and it's totally fine to present them the way the authors did, but to help the reader to understand what and why was performed, it would be good to at least mention that the experiment did not go as expected.
5. In the CtAtm1_{inw-opn}/cluster structure, an additional density is visible, which is of sufficient size to accommodate [2Fe-2S](GS)₄ cluster, however, due to the low resolution, its identity cannot be confirmed. I do agree with the authors that the most likely origin of this extra density is [2Fe-2S](GS)₄ cluster, but since all the story focuses on [2Fe-2S] transport, I think the authors should use other methods like mutagenesis of interacting residues to validate this.
6. I do like that the authors honestly state that the occlusion of [2Fe-2S] did not work and, therefore, they needed to change the cargo to GSSG, which was also, unfortunately, not visible in the occluded structure. I really like that the authors manage to take all the "failed" experiments and still make a quite nice story out of them, providing important insights into the selectivity of the ligand binding pocket. However, without clearly seeing the [2Fe-2S] in any of the structures, I find the title "Structures of [2Fe-2S] Cluster Export by Mitochondrial Amt1" a bit misleading and also grammatically not fully correct. In addition, there is a typo in the current title (should be Atm1 instead of Amt1).

Minor details:

1. While Atm1 abbreviation is commonly used in ABC transporter field, I think the authors should still provide the explanation of the abbreviation for the broader community the first time they mention it.
2. Fig. 4: The grey density for ATP is very hard to see, the authors should change the colour of the density to make it stand out more. The same is the case in Supplementary Fig. 5E – it would be much clearer if the density for ATP would be highlighted in another colour.
3. The authors should include at least one representative micrograph into the supplements.
4. In the methods section, the authors write that "For the CtAtm1occl conformation, the same E603Q form in POCP nanodiscs was incubated with 1 mM GSSG, 1 mM AMP-PNP and 1 mM MgCl₂", however, in the main text they write: "To generate an occluded configuration we again exploited the E603Q form, but this time we supplemented the sample with ATP and cargo". So what was actually used to obtain CtAtm1occl conformation, ATP or AMP-PNP?

We thank the reviewers for the evaluation and for the helpful suggestions and comments to improve the manuscript. We have now addressed all comments and amended the manuscript as outlined below, also providing considerable amounts of additional data. Remarks and questions from the reviewers are shown in black. Our responses are shown in red.

Reviewer #1 (Remarks to the Author):

The manuscript presents new structures of an Atm1 homologue, obtained by cryoEM. This is an excellent technical achievement, however the results are very similar to those published in 2014 by the groups of Lill (yeast Atm1) and Rees (bacterial Atm1 homologue). The discussion of important residues and comparison of the new structures with the previous structures is very informative.

The title maybe misleading? The words "Structures of Fe-S cluster export" suggests movement, but the data mainly consist of Cryo-EM structures, which are static. Moreover, there is insufficient resolution to be certain about the Fe-S cluster in one structure which supposedly has it, whereas the other structures do not have Fe-S bound (but GSSG).

Answer 1: We have now revised the title. The structural data are conclusive on the presence of a Fe-S cluster. Firstly, the cluster density has only been captured when the cluster is supplemented to the sample. Control measurements without cluster supplementation to the sample do not show the additional cluster density (as for example in the CtAtm1^{inw-opn} structure). Secondly, the cluster density appears also at a high signal to noise ratio. At contour level 1.3 the protein is almost invisible in the cryo-EM map, but a clear peak is still visible for the cluster (Supplementary Fig. 12). Thus, the peak cannot represent protein or GSH/GSSG. Thirdly, with lower signal to noise ratio (contour level 0.4) at which sidechains appear in the cryo-EM density, the volume of the plate shape density is around 3546 Å³, which is better explained by [2Fe-2S](GS)₄ than by GSH/GSSG. Moreover, the shape of the density feature is plate shaped (and not spherical, Fig. 2a), as can be expected by the cluster. Fourthly, at contour level 1.3 (when the protein is almost invisible), the volume of the remaining cryo-EM peak is square shaped and with an approximate volume of 150 Å³, which even at this contour level would be sufficiently large for a 2Fe-2S cluster. Collectively, these observations clearly support the presence of the cluster at the proposed location. We note we have been unable to model the cluster, which hints at some flexibility of the cluster binding in the CtAtm1^{inw-opn/cluster} conformation, which otherwise would be likely to further extend the cryo-EM density feature, stretching further towards the protein.

We also provide new functional data (Fig. 1c and Fig. 2d,e), which show the cluster stimulates wild-type CtAtm1 ATPase activity, and that the cluster is more potent in this respect than GSH and GSSG (Fig. 1c). UV-VIS spectroscopy of the protein fraction following desalting chromatography of wild-type CtAtm1 mixed with the cluster exhibits features at around 320 and 415 nm, respectively (Fig. 2e). These features represent established fingerprints of [2Fe-2S] clusters (Chem Commun (Camb). 2016 Nov 10;52(92):13456-13459).

Control experiments conducted in the absence of the cluster do not show these characteristics. Equivalently, mutations of residues of CtAtm1 that line the cluster in CtAtm1^{inw-opn/cluster} reduce the features. These are all observations that support complex formation in wild-type CtAtm1, decreased binding for the mutant forms, and absence of binding when the cluster has not been supplemented. We have also performed ICP-MS and the results show the purified WT-cluster complex gives higher iron content than the mutants.

Abstract “these clusters are primarily biosynthesized in mitochondria and then exported to the cytoplasm using ABCB7-type proteins” – the evidence in the literature for this is not strong, by all means, but this is not reflected in the wording here. Note, in the Introduction, the authors state “Even the type of cargo for the eukaryotic homologs has not been clearly elucidated.”

We have now modified the sentence in the abstract.

Key papers on this topic reporting export of GSSG and persulfide are ignored. For example, Schaedler et al 2014 JBC, Pandey et al 2019 JBC. These papers should be summarized and cited in the Introduction, to balance self-citations 11 – 13 and 21.

The introduction has been expanded and the indicated references have now been included in the manuscript and discussed.

Fig 1b. Are the E>Q mutant proteins expressed correctly, e.g. at comparable levels with wild type protein and not affected in their folding? The ATPase activity of the mutant proteins without cargo is unusually low. Every protein is different in this respect, and ‘classical’ amino acid substitutions that work well for one ABC transporter, may not work for an orthologue from another species. Please add information on the typical protein yield per liter bacterial culture and show Coomassie-stained gels of the purified proteins. What are ATPase activities with cargo, e.g. GSSG? No data are given for this.

We have now provided size exclusion chromatography (SEC) profiles and Coomassie-stained SDS-PAGE analyses of all tested CtAtm1 forms in Supplementary Fig. 20. The analysis suggests the E603Q form is well-behaved similarly to wild-type and new introduced mutant forms, with maintained yields and a homogenous SEC peak. The typical yield is around 5 mg purified protein (following SEC) from 1 L of cells cultured in Terrific Broth (TB) medium, with modestly lower yields for certain mutant forms, including E603Q. Collectively, these data suggest all the examined CtAtm1 forms are correctly folded.

We have now also introduced functional data in the presence of GSH, GSSG and the cluster, respectively, as compared to CtAtm1 without supplementation (Fig. 1c). Clearly, cluster stimulates the activity stronger than GSH or GSSH, hinting that a cluster is at least an equally potent cargo as GSH and GSSG. Supplementary Fig. 2c shows robust activity without supplementation of GSH, GSSG or the cluster, which is inhibited by the E603Q substitution. E603Q is also not functional in the presence of the cluster (Fig. 1c), suggesting the stimulatory effect of the cluster is specific to functional CtAtm1.

We note CtAtm1 forms homodimer, and that a single E603Q mutation hence affects both the catalytic glutamates, explaining why the mutant has very low ATPase activity (comparable to the buffer control without protein). This is consistent with the fact that NaAtm1 and AtAtm3 almost completely lose activity with the E2Q mutant.

The Fe-S/GSH cargo is deduced from modelling and docking studies. Evidence for binding of the cargo, and that this is physiological, should be provided. For example by mutagenesis of the amino acids that are modelled to interact with the ‘disc-like structure’. A UV-vis spectrum of the purified protein binds Fe-S/GSH should be shown.

See above regarding interpretation of the cryo-EM data, as well as new functional data provided in Fig. 1c and Fig. 2d,e. Fig. 1c indicates that the cluster stimulates the CtAtm1 more than GSH or GSSH. The new UV-VIS data in Fig. 2d, suggest binding of a [2Fe-2S] to CtAtm1, and that this binding is decreased by mutations of the residues that interact with the cluster in the CtAtm1^{inw-opn/cluster} structure. We also have measured the iron content with purified protein cluster complex sample and mutants, the results show the mutants has lower iron than the WT.

Reviewer #2 (Remarks to the Author):

This was an excellent and exciting paper overall; the ability of the authors to obtain a high-resolution cryo-EM structure of an Atm1, with cargo bound, is remarkable and highly significant. The authors used very creative means of preventing the translocation of cargo (inactive mutant protein and derivative of ATP that cannot be hydrolyzed) in obtaining their structures.

We thank the reviewer for these positive words regarding our work.

Nevertheless, I have a few concerns. My major concern is the need to clarify the degree to which the authors are certain that the density observed upon adding the cargo is in fact an $[2Fe_2S](GSH)_4$ cluster.

See Answer 1 above (response to reviewer 1).

They state unambiguously in the abstract and discussion section that this has been "demonstrated". However, they seem to be more cautious in the description in the Results section and I wasn't able to discern the strength of their evidence. There is a long history to this. On page 2, the authors state that the $[2Fe_2S](GSH)_4$ cluster "is of interest because such complexes are stable in aqueous media thereby permitting $[2Fe_2S]$ cluster to be mobilized in the cell without a protein scaffold (Ref 12 – 15)". However, many researchers in this field are skeptical that $[2Fe_2S](GSH)_4$ clusters can exist stable in aqueous solution. All of the references supporting this statement are from the same lab, as no other lab has confirmed this. I am agnostic on this issue, but am certain that all of the doubters would be silenced immediately if the authors would publish two strong pieces of evidence including: a) a full high-resolution ESI-MS spectrum of the cluster after being incubated in aqueous buffer for a few days, showing the predicted masses including the expected isotopomers in the expected intensity ratios; and b) Mossbauer spectra of the same, showing a clean quadrupole doublet with parameters of a $[2Fe_2S]^{2+}$ cluster and with no cluster degradation after incubation.

From the response in Answer 1 (above, response to reviewer 1) it should be clear that there is no doubt the cluster binds to CtAtm1 in the CtAtm1^{inw-opn/cluster} structure. Furthermore, the cluster stimulates the function more than GSH and GSSG, respectively. The mass spectrometric and Mossbauer support for the glutathione-complexed cluster suggested by the reviewer have already been published. In particular, the reviewer is directed to the following work and respective supplementary information in those publications: J Am Chem Soc. 2012 Jul 4;134(26):10745-8, and Chem Commun (Camb). 2013 Jul 18;49(56):6313-5.

With respect to the current data, I couldn't tell whether the authors were able to establish that the cargo definitely contains iron. Is the size and shape of the cargo similar to what would be expected if it were due to this cluster? This issue should take "center stage" in the current paper.

From the expanded analysis of the cryo-EM data in Answer 1 (above, response to reviewer 1), it is clear that the observed density represents a cluster with intact $[2Fe_2S]$. This is supported by the size and shape of the cryo-EM density detected also by the new functional data, including Fig. 2d. We also have performed ICP-MS measurements to assess the purified protein and cluster complex sample and the results show there is iron present in the protein-cluster complex sample (Fig. 2e).

The only other concern was that the history of the Atm1 story was not well developed in the introduction. The recent papers from the Pain/Dancis lab on this issue (Pandey et al) were not mentioned but should be. Relevant work by Janneke Balk on glutathione trisulfide should be mentioned as well. Finally, the title of the article should read Atm1 not Amt1. They should also mention in the title and abstract the source of Atm1 used in their study.

The introduction has been expanded and the indicated references have now been included in the manuscript and discussed. The title has been revised. The source of the Atm1 in the study has been clarified. We refrain from specifying the source due to space constraints imposed by the journal.

Reviewer #3 (Remarks to the Author):

In this manuscript, Li et al. employ cryo-EM to investigate the mechanism of [2Fe-2S] export by eukaryotic ABC transporter Atm1. As their model system, the authors use nanodisc-reconstituted CtAtm1 from thermophilic fungus. Utilizing ATP or AMP-PNP trapping, the authors report a series of structures, providing important insights into the transport mechanism and selectivity of the cargo. I have no doubt that this manuscript would be of high interest to many researchers in ABC transporter field, however, I still have some concerns that should be addressed.

We thank reviewer for the favourable evaluation of the work.

1. The authors obtained the structure of wild-type CtAtm1_{inw-opn} in the presence of ATP, however, in the methods section there is no information on how exactly this particular experiment was performed: what concentration of ATP and Mg²⁺ was used and at which temperature and for how long the sample was incubated prior to grid preparation. The authors should add the missing information.

We thank the reviewer for pinpointing these shortages. We have now provided more experimental details regarding the sample preparation. The final concentration used for the grid preparation were 1 mM ATP and 2 mM Mg²⁺. The wild-type sample was incubated on ice for 30 min prior to the grid preparation.

2. On page 5, the authors write: "... the resolution of the NBDs is somewhat lower, and hence we are unable to confirm the absence of nucleotide binding". Lower resolution at the NBDs is not surprising due to their flexibility, but looking at the processing scheme in Supplementary Fig. 4, it does not seem that the authors tried to improve the densities at the NBDs, for example, by performing local refinement with or without symmetry expansion. I would recommend to do that, because I am quite surprised that the authors expect the absence of nucleotides even though they added ATP. I would expect that the ATP should have been hydrolyzed, but ADP is still bound in the structure - in such a case, I would also say that the authors statement that it is "a pre-translocation arrangement" is debatable, but it is really hard to judge this without having any details on how this experiment was actually performed, as already mentioned in my first comment.

We have now performed further data processing using the 3D viability function in cryosparc. The results indicate that while the membrane domain of Atm1 is almost identical, the NBD domain shows a high degree of heterogeneity, including both side movement and rotation, which could be clearly seen in the attached movies. Because there is no clear and dominant conformation for the NBDs, we could not further improve the density of the NBDs sufficiently to identify the nucleotides (if present). We suggest that NBD movement may be an intrinsic feature of this protein without presence of cargo. As indicated in the text, a similar conformation of NaAtm1 has been interpreted as a pre-translocation state previously.

3. In addition, while the authors report only one final structure from this dataset, Supplementary Fig. 4 displays several different conformations with variable degree of NBD separation (best visible in 2D classes). Did the authors look into this and tried to refine the different conformations separately instead of just going for a single highest resolution structure?

We thank the reviewer and her/his deep understanding of cryo-EM data quality/processing. As indicated above, we have executed further data processing efforts, and we have indeed identified a small subset of the wild-type cryo-EM data (about 17.9 % of the total particles) that represent a similar conformation as the inward-facing partially occluded structure in the [2Fe-2S] cluster dataset (see revised Supplementary Fig. 4). However, in contrast to the E603Q+AMPPNP+cluster data, this particular state has no 2xGSH densities in the pocket (but it should be mentioned that

the resolution is lower, 4.2 Å). Thus, it is likely that the GSH density in the inward-facing partially occluded structure (CtAtm1^{inw-opn-occl}) is indeed coming from the added cluster rather than coming along from the purified protein itself (as we propose in the manuscript). This is in fact also indirect evidence that the cluster can serve as a cargo for Atm1 as proposed in the mechanistic scheme in Fig. 5.

4. It is also not clear to me, why for trying to capture the structure with the cargo, the authors utilize both the EQ mutant and AMP-PNP. Normally, one would go for either EQ mutant with ATP or wild-type with AMP-PNP, but the authors chose the option with both modifications - is there any specific reason for that? Since the authors didn't obtain any structures with dimerized NBDs from this dataset, AMP-PNP trapping seems to simply have not worked and the same structures would have been probably obtained without addition of any nucleotides. Did the authors try to obtain such data? Even though the trapping didn't work, the structures are still valuable and it's totally fine to present them the way the authors did, but to help the reader to understand what and why was performed, it would be good to at least mention that the experiment did not go as expected.

We thank reviewer for these remarks, and we agree with the proposed strategy of the reviewer. In fact, we also tried capturing a cargo-bound structure using wild-type sample with cluster and AMP-PNP, however, we only obtained the inward open (apo) and the inward-open bound with cargo (shape-like density bound) conformations, and at relatively low resolution, compared to the structures presented in the manuscript (now included in Supplementary Fig. 9). We also attempted using the E603Q form and ATP, but the sample precipitated, preventing cryo-EM efforts (included in the revised version). These points are now all mentioned in the manuscript. This is why we tried the E603Q form supplied with AMPPNP instead of ATP to capture the cargo bound states. Notably, the cluster feature was not present when data was collected using sample in which the cluster was replaced with GSSG (data collection was stopped as the feature was not appearing in contrast to the data collections in the presence of AMPPNP and cluster, data not shown).

5. In the CtAtm1^{inw-opn/cluster} structure, an additional density is visible, which is of sufficient size to accommodate [2Fe-2S](GS)₄ cluster, however, due to the low resolution, its identity cannot be confirmed. I do agree with the authors that the most likely origin of this extra density is [2Fe-2S](GS)₄ cluster, but since all the story focuses on [2Fe-2S] transport, I think the authors should use other methods like mutagenesis of interacting residues to validate this.

From the response in Answer 1 (above, response to reviewer 1) it should be clear that there is no doubt the cluster binds to CtAtm1 in the CtAtm1^{inw-opn/cluster} structure, and that the observed density represents a cluster with intact [2Fe-2S]. Furthermore, the cluster stimulates the function more than GSH and GSSG, respectively (Fig. 1c). Binding is also supported by new UV-VIS data, Fig. 2d, suggesting that mutations of residues that interact with the cluster in the CtAtm1^{inw-opn/cluster} structure reduce the binding. We also have performed the ICP-MS to measure the purified protein and cluster complex sample and the results show there is iron present in the protein-cluster complex sample and that the amount of bound iron is decreased when the residues exposed to cluster are mutated (Fig. 2e).

6. I do like that the authors honestly state that the occlusion of [2Fe-2S] did not work and, therefore, they needed to change the cargo to GSSG, which was also, unfortunately, not visible in the occluded structure. I really like that the authors manage to take all the "failed" experiments and still make a quite nice story out of them, providing important insights into the selectivity of the ligand binding pocket. However, without clearly seeing the [2Fe-2S] in any of the structures, I find the title "Structures of [2Fe-2S] Cluster Export by Mitochondrial Amt1" a bit misleading and also grammatically not fully correct. In addition, there is a typo in the current title (should be Atm1 instead of Amt1).

Based on the above-mentioned arguments (see also Answer 1, response to reviewer 1), we are convinced that a cluster including [2Fe-2S] binds to CtAtm1 in the CtAtm1^{inw-opn/cluster} structure. We have modified the titled and corrected the error.

Minor details:

1. While Atm1 abbreviation is commonly used in ABC transporter field, I think the authors should still provide the explanation of the abbreviation for the broader community the first time they mention it.

We have now included the explanation of the abbreviation in the revised version.

2. Fig. 4: The grey density for ATP is very hard to see, the authors should change the colour of the density to make it stand out more. The same is the case in Supplementary Fig. 5E – it would be much clearer if the density for ATP would be highlighted in another colour.

We have now changed the density colour for ATP in both Fig. 4 and Supplementary Fig. 5.

3. The authors should include at least one representative micrograph into the supplements.

We have now added micrographs for each data set.

4. In the methods section, the authors write that "For the CtAtm1^{occl} conformation, the same E603Q form in POCP nanodiscs was incubated with 1 mM GSSG, 1 mM AMP-PNP and 1 mM MgCl₂", however, in the main text they write: "To generate an occluded configuration we again exploited the E603Q form, but this time we supplemented the sample with ATP and cargo". So what was actually used to obtain CtAtm1^{occl} conformation, ATP or AMP-PNP?

The old method section was incorrect. We obtained the CtAtm1^{occl/ATP} conformation with E603Q in the presence of 1 mM ATP and 1 mM MgCl₂, and supplied with 1 mM GSSG. We have corrected this in the revised version.

REVIEWERS' COMMENTS

Reviewer #1 (Remarks to the Author):

The change of the title from "Structures of [2Fe-2S] Cluster Export by Mitochondrial Amt1 [sic]" to "Structural basis of [2Fe-2S] Cluster Export by Mitochondrial Atm1" does not address the point that the word "export" is misleading because it suggests movement, whereas structures are static. In the ATPase stimulation data in Fig 1C, does the 20 μ M cluster contain 80 μ M glutathione (4 ligands)? If so, and the cluster falls apart in the assay, the concentration of glutathione is 4 times higher than GSH and 2 times higher than GSSG.

The authors have provided more data on the Q603E mutant, which is helpful and satisfies my query.

For the changed section of the Abstract "... the molecular mechanism for how the clusters are translocated from mitochondria" has the export of Fe-S clusters as starting point, however, others have proposed that only persulfide is exported. Thus, the new wording does not address the point that assumption are made and that the existing literature is not sufficiently taken into account. Even worse, in the sentence "...glutathione-bound iron sulfur clusters, [2Fe-2S](GS₄), represent the most likely cargo to provide clusters for cytosolic proteins, consistent with the observation of an "Fe-S" cargo that facilitates CIA-type cluster assembly (Ref 24,25)", but Ref 24 does not show or even suggest this at all.

For assessment of the densities that may or may not indicate an Fe-S cluster, I will leave this to the other two reviewers, as I am not a crystallography expert.

Reviewer #2 (Remarks to the Author):

Main comment: I recommend that the paper be accepted for publication as it will be of great interest to the cellular iron trafficking field. However, I strongly recommend that the authors "tone down" their claim to have unambiguously and definitively identified the cargo as a [2Fe-2S] cluster coordinated to glutathione. Unfortunately, they simply don't have the resolution in their data to draw that conclusion. The current title "Structural basis of [2Fe-2S] cluster export by Mitochondrial Atm1" still implies that they have proved this. In the abstract, the authors state "We demonstrate that CtAtm1 accepts delivery from glutathione-complexed iron-sulfur clusters" - but "demonstrate" is too strong. In the summary on page 16 they state that "a glutathione-complexed [2Fe-2S] cluster is identified" as bound cargo, without any ambiguity. In contrast, the tone in the Results and Discussion section is far more nuanced and scientifically balanced. On page 4, they state that GSH-[2Fe₂S] clusters are "the most likely cargo" (I agree). On page 8 they state that "a large plate-shaped additional feature" that is significantly larger than GSH or GSSG, and is consistent with the cluster (OK, but they should give the volume of the cluster - only the volume of the cargo is 3500 Å³ is given). On page 14, they state "Our data is inconclusive regarding the nature of the bound cargo because only parts of the cluster are detected." All of these statements sound more reasonable to me and are a truer reflection of their data. In brief, the authors continue to overstate their results in the title, abstract, and summary. I had the same complaint in my first review, and little was changed in the revised version.

Other comments: Page 12: the inner gate is too small to allow "transfer of the entire cluster, and rather hints that the separate pathways (which ones are these) handle one part of the cluster in the export process, preparing the cargo for the export in the anticipated outward-facing conformation". I'm not sure what this means - the cluster is divided and sent through two different pathways?? Then they reassure us that the "pocket is sufficiently large to accommodate and entire" cluster.

Page 21: Fig. 2e; there is about 2.8 - 3.6 Fe/atm1 with uncertainties so low as to exclude the value 2 Fe/atm1. Technically, their result is incompatible with the presence of a [2Fe-2S] cluster.

Page 21: Fig 2d; The UV-vis spectrum is similar to those of [2Fe-2S] clusters but the characteristic features are less apparent/intense. If the absorbance at say 420 nm is divided by half the Fe concentration, is the resulting extinction coefficient similar to that expected for [2Fe-2S] clusters?

Reviewer #3 (Remarks to the Author):

The authors have addressed all my concerns and greatly improved the manuscript through their revision, providing more details and more certainty, especially, regarding [2Fe-2S](GS)₄ cluster binding in the CtAtm1inw-opn/cluster structure. It is a great story and I can only recommend it for publication.

We thank the reviewers for further evaluating and for the helpful suggestions and comments to improve the manuscript. We have now addressed all comments and amended the manuscript as outlined below. Our responses are shown in red. The changes are high-lighted in yellow in the manuscript files.

Reviewer #1 (Remarks to the Author):

The change of the title from "Structures of [2Fe-2S] Cluster Export by Mitochondrial Amt1 [sic]" to "Structural basis of [2Fe-2S] Cluster Export by Mitochondrial Amt1" does not address the point that the word "export" is misleading because it suggests movement, whereas structures are static.

We have changed the title to "Structures of Amt1 provide insight into [2Fe-2S] cluster export from mitochondria".

In the ATPase stimulation data in Fig 1C, does the 20 uM cluster contain 80 uM glutathione (4 ligands)? If so, and the cluster falls apart in the assay, the concentration of glutathione is 4 times higher than GSH and 2 times higher than GSSG.

Theoretically, the glutathione-complexed cluster contains 4 glutathione and 1 [2Fe-2S] cluster center, but it should be considered as 1 compound and should not fall apart into 4 glutathione. However, we have now tested the ATPase activity in the presence of 80 μ M GSH or 40 μ M GSSG individually. The results show the ATPase activity is increased further in the presence of 80 μ M GSH, and similarly with 40 μ M GSSG simulation. However, the activity remains lower than in the presence of 20 μ M of the cluster.

The authors have provided more data on the E603Q mutant, which is helpful and satisfies my query.

We thank the comments regarding the new E603Q data.

For the changed section of the Abstract "... the molecular mechanism for how the clusters are translocated from mitochondria" has the export of Fe-S clusters as starting point, however, others have proposed that only persulfide is exported. Thus, the new wording does not address the point that assumption are made and that the existing literature is not sufficiently taken into account.

We have now further revised the abstract.

Even worse, in the sentence "...glutathione-bound iron sulfur clusters, [2Fe-2S](GS4), represent the most likely cargo to provide clusters for cytosolic proteins, consistent with the observation of an "Fe-S" cargo that facilitates CIA-type cluster assembly (Ref 24,25)", but Ref 24 does not show or even suggest this at all.

We have rewritten this section to make it clearer. Ref24 showed that GSH depletion affects the activity of cytosolic iron-sulphur containing proteins, which is possible as GSH serves as a cofactor that provides Fe-S clusters for the maturation of cytosolic iron-sulphur protein.

For assessment of the densities that may or may not indicate an Fe-S cluster, I will leave this to the other two reviewers, as I am not a crystallography expert. Based on the data presented in the manuscript, we are convinced the plate shape cryo-EM density represents a glutathione-complexed Fe-S cluster.

Reviewer #2 (Remarks to the Author):

Main comment: I recommend that the paper be accepted for publication as it will be of great interest to the cellular iron trafficking field. However, I strongly recommend that the authors "tone down" their claim to have unambiguously and definitively identified the cargo as a [2Fe-2S] cluster coordinated to glutathione. Unfortunately, they simply don't have the resolution in their data to draw that conclusion. The current title "Structural basis of [2Fe-2S] cluster export by Mitochondrial Atm1" still implies that they have proved this.

We thank the reviewer for recommending our paper for publication and for the suggestions for further improving the manuscript. We have now revised the title.

In the abstract, the authors state "We demonstrate that CtAtm1 accepts delivery from glutathione-complexes iron-sulfur clusters" - but "demonstrate" is too strong. In the summary on page 16 they state that "a glutathione-complexed [2Fe-2S] cluster is identified" as bound cargo, without any ambiguity. In contrast, the tone in the Results and Discussion section is far more nuanced and scientifically balanced.

We have revised the abstract and summary sections.

On page 4, they state that GSH-[2Fe2S] clusters are “the most likely cargo” (I agree). On page 8 they state that “a large plate-shaped additional feature” that is significantly larger than GSH or GSSG, and is consistent with the cluster (OK, but they should give the volume of the cluster – only the volume of the cargo is 3500 Å³ is given).

The volume of the unsharp cargo density is around 3500 Å³, as assessed using the ‘volume blob measurement’ tool in the software chimera. The molecular weight of the glutathione-coordinated [2Fe-2S] is 1.4 kD, so the estimated volume of the glutathione-coordinated [2Fe-2S] is around 1700 Å³ (ref, PMID: 19495910). The volume of the docking model of the cluster in our structure is calculated around 1100 Å³. Therefore, the plate shaped additional density is sufficiently large to accommodate the entire cluster. The 370 and 740 Å³ volumes of GSH and GSSG have been added to the manuscript.

On page 14, they state “Our data is inconclusive regarding the nature of the bound cargo because only parts of the cluster are detected.” All of these statements sound more reasonable to me and are a truer reflection of their data. In brief, the authors continue to overstate their results in the title, abstract, and summary. I had the same complaint in my first review, and little was changed in the revised version.

We have revised the title, abstract and the summary.

Other comments: Page 12: the inner gate is too small to allow “transfer of the entire cluster, and rather hints that the separate pathways (which ones are these) handle one part of the cluster in the export process, preparing the cargo for the export in the anticipated outward-facing conformation”. I’m not sure what this means – the cluster is divided and sent through two different pathways?? Then they reassure us that the “pocket is sufficiently large to accommodate and entire” cluster.

We have revised this section. The occluded state we have captured likely represents an early occluded conformation. Based on the outward facing structure of AtAtm3, in which TM6 is straightened, we believe there should exist another occluded intermediate, in which TM6 is straightened to create a large cavity to accumulate the cargo. This would prepare the cargo for export in the anticipated outward-facing conformation with TM6 maintained straightened. This will need further investigation.

Page 21: Fig. 2e; there is about 2.8 – 3.6 Fe/atm1 with uncertainties so low as to exclude the value 2 Fe/atm1. Technically, their result is incompatible with the presence of a [2Fe-2S] cluster.

The indicated stoichiometry is on a molar ratio, so one would expect 2 Fe for 1 dimer, or 2 Fe for 2 monomers. The figure denotes the molar ratio of Fe versus monomer. Hence, the molar ratio is unexpectedly high. One possibility is that the

cluster binds in more than 1 nearly overlapping site, since the volume we detected is 3500 \AA^3 and the cluster itself is around 1700 \AA^3 . In this regard, we cannot exclude the presence of a low-affinity (or adventitiously bound) [2Fe-2S] cluster binding site(s) beyond the one determined in the cryo-EM structure, which could not be completely removed by the desalting purification step, thus resulting in a higher-than-expected stoichiometry. Another possible contributing factor is that the desalting column (a 5 mL Hi Trap from Cytiva featuring the Sephadex G-25 resin) would have insufficient resolution to separate with a 100 % efficiency holo Amt1 from excess [2Fe-2S] free cluster, thus leading to a minor contamination of unbound cluster in the Amt1 (and mutants) fractions. These aspects, together with a potential minor underestimation of the protein concentration by Abs_{280} using the Nanodrop instrument, may have contributed to the determination of higher-than-expected stoichiometries. Nevertheless, these measurements allow to confirm the presence of bound cluster to Amt1 as well as compare the relative binding ability of mutants.

Page 21: Fig 2d; The UV-vis spectrum is similar to those of [2Fe-2S] clusters but the characteristic features are less apparent/intense. If the absorbance at say 420 nm is divided by half the Fe concentration, is the resulting extinction coefficient similar to that expected for [2Fe-2S] clusters?

We agree the characteristic [2Fe-2S] features are less intense in our Amt1 samples. Based on the absorbance of cluster-bound Amt1, the extinction coefficients at 425 nm and 330 nm ($\sim 3570 \text{ M}^{-1}\text{cm}^{-1}$ and $8090 \text{ M}^{-1}\text{cm}^{-1}$, respectively) are consistent with prior reports of the absorbance spectra, the extinction coefficients, and A_{330}/A_{425} ratio for glutathione-complexed [2Fe-2S] cluster (Ref 17 in the main text). Observed minor deviations are expected based on changes in polarity of the protein-bound cluster and perturbation of the binding geometry for ligated glutathiones.

Reviewer #3 (Remarks to the Author):

The authors have addressed all my concerns and greatly improved the manuscript through their revision, providing more details and more certainty, especially, regarding [2Fe-2S](GS)₄ cluster binding in the CtAmt1_{inw-opn}/cluster structure. It is a great story and I can only recommend it for publication.

We thank the reviewer for recommending our work for publication.